EMBO
Molecular Medicine

# Endothelial Notch signaling controls insulin transport in muscle

Sana S Hasan[1,†], Markus Jabs[1,†], Jacqueline Taylor[1,2], Lena Wiedmann[1,2], Thomas Leibing[3,4], Viola Nordström[5], Giuseppina Federico[5], Leticia P Roma[6], Christopher Carlein[6], Gretchen Wolff[7], Bilgen Ekim-Üstünel[7], Maik Brune[8], Iris Moll[1], Fabian Tetzlaff[1], Hermann-Josef Gröne[5,9], Thomas Fleming[8], Cyrill Géraud[3,4,10], Stephan Herzig[7,8] [ID], Peter P Nawroth[7,8] [ID] & Andreas Fischer[1,8,10,*] [ID]

## Abstract

The role of the endothelium is not just limited to acting as an inert barrier for facilitating blood transport. Endothelial cells (ECs), through expression of a repertoire of angiocrine molecules, regulate metabolic demands in an organ-specific manner. Insulin flux across the endothelium to muscle cells is a rate-limiting process influencing insulin-mediated lowering of blood glucose. Here, we demonstrate that Notch signaling in ECs regulates insulin transport to muscle. Notch signaling activity was higher in ECs isolated from obese mice compared to non-obese. Sustained Notch signaling in ECs lowered insulin sensitivity and increased blood glucose levels. On the contrary, EC-specific inhibition of Notch signaling increased insulin sensitivity and improved glucose tolerance and glucose uptake in muscle in a high-fat diet-induced insulin resistance model. This was associated with increased transcription of *Cav1*, *Cav2*, and *Cavin1*, higher number of caveolae in ECs, and insulin uptake rates, as well as increased microvessel density. These data imply that Notch signaling in the endothelium actively controls insulin sensitivity and glucose homeostasis and may therefore represent a therapeutic target for diabetes.

**Keywords** caveolae; endothelial cell; insulin transport; muscle; Notch signaling

**Subject Categories** Metabolism; Vascular Biology & Angiogenesis

## Introduction

The inner lining of blood vessels is composed of endothelial cells (ECs) which provide an anti-thrombotic surface for transportation of blood to distal organs. In addition, ECs also regulate tissue regeneration, stem cell renewal and differentiation, and tumor progression through paracrine (angiocrine) interactions in an organ-specific manner (Rafii *et al*, 2016; Augustin & Koh, 2017). As ECs form a barrier between blood and all other cell types in the human body, they possess a unique spatial advantage to act as gatekeepers and maintain metabolic homeostasis by controlling the access of nutrients and hormones from blood to the surrounding tissue.

ECs have different characteristics, which contribute to organ-specific vascular beds. For example, fenestrated endothelium containing pores allow for rapid exchange of water and small solutes in kidneys while sinusoidal endothelium form gaps that facilitate passive transport of larger molecules to hepatocytes in liver. On the other hand, continuous endothelium, e.g., in muscle, brain, and skin, provides a restrictive transcellular flux of nutrients and hormones (Aird, 2007; Augustin & Koh, 2017).

Circulating hormones like insulin must cross the continuous endothelium to reach its target cells. It has been suggested that insulin transport across the endothelium to muscle tissue (the major site of insulin-mediated glucose uptake) is the rate-limiting step in insulin-mediated lowering of blood glucose (Yang *et al*, 1989). Although a model of non-saturable fluid-phase insulin transport has recently been described (Williams *et al*, 2018), there is strong evidence that the rapid insulin transport to muscle cells occurs through ECs and requires the presence of insulin receptors on ECs

1 Division Vascular Signaling and Cancer (A270), German Cancer Research Center (DKFZ), Heidelberg, Germany
2 Faculty of Biosciences, University of Heidelberg, Heidelberg, Germany
3 Department of Dermatology, Venereology, and Allergology, University Medical Center and Medical Faculty Mannheim, Heidelberg University, Mannheim, Germany
4 Section of Clinical and Molecular Dermatology, Medical Faculty Mannheim, Heidelberg University, Mannheim, Germany
5 Division of Cellular and Molecular Pathology, German Cancer Research Center (DKFZ), Heidelberg, Germany
6 Biophysics Department, Center for Human and Molecular Biology (ZHMB), Saarland University, Homburg, Germany
7 Institute for Diabetes and Cancer (IDC) and Joint Heidelberg-IDC Translational Diabetes Program, Helmholtz Center Munich, Neuherberg, Germany
8 Department of Medicine I and Clinical Chemistry, University Hospital of Heidelberg, Heidelberg, Germany
9 Institute of Pharmacology, Philipps University of Marburg, Marburg, Germany
10 European Center for Angioscience, Medical Faculty Mannheim, Heidelberg University, Mannheim, Germany
  *Corresponding author. Tel: +49 1622 1424150; E-mail: a.fischer@dkfz.de
  †These authors contributed equally to this work

(Chernick *et al*, 1987; Bar *et al*, 1988; Baura *et al*, 1993; Konishi *et al*, 2017).

The mechanism by which insulin crosses the continuous endothelium is critical to understand insulin action and insulin resistance. Trans-endothelial insulin transport varies among vascular beds and requires either clathrin-coated vesicles or caveolae (Wang *et al*, 2011; Azizi *et al*, 2015). Caveolae are specialized lipid rafts whose formation is dependent on proteins of the Caveolin (Cav) and Cavin families. Insulin binds to its receptor on ECs, is internalized by caveolae (Wang *et al*, 2006; Barrett *et al*, 2009), and is released at the basolateral EC membrane, where it diffuses in the interstitium and activates insulin-mediated pathways in muscle cells (Barrett *et al*, 2011).

The caveolar protein Cav1 is essential for the formation of caveolae irrespective of the cell type (Sowa, 2012). *In vitro*, overexpression of Cav1 increases EC insulin transport rates, whereas a reduction in Cav1 expression impairs insulin flux (Wang *et al*, 2011). However, *in vivo* the situation is more complex. Inactivation of the *Cav1* gene leads to lower numbers of caveolae, but this also opens paracellular routes to compensate for the impaired transcellular flux (Schubert *et al*, 2002). Likewise, overexpression of Cav1 alone does not increase the abundance of caveolae, implying that additional proteins are involved in the control of caveolae numbers (Razani *et al*, 2002; Bauer *et al*, 2005). The muscular endothelium expresses multiple proteins linked to caveolae formation, including Cav2, which interacts with Cav1, and Cavin1, an adaptor protein crucial for caveolae stabilization (Hansen *et al*, 2013).

Even though in recent years extensive research has emerged into mechanisms of organ-specific vascular development, the knowledge about signaling pathways that control endothelial transport of hormones and nutrients is still very preliminary. Very recently, we have shown that endothelial-specific Notch signaling is required for fatty acid transport to muscle cells (Jabs *et al*, 2018). Notch signaling cascade is a juxtacrine communication system that requires the binding of ligands from the Delta-like (Dll) and Jagged family to Notch receptors on adjacent cells. This in turn induces receptor cleavage and leads to translocation of Notch intracellular domain (NICD) to the nucleus where it interacts with transcriptional co-activators like Mastermind like-1 (MAML1) and Rbp-jκ on gene promoters (Kopan & Ilagan, 2009). The activity of Notch signaling is influenced by the nutritional status and certain plasma metabolites and has been shown to be crucial for controlling glucose metabolism in hepatocytes and adipocytes (Pajvani *et al*, 2011, 2013; Bi *et al*, 2014; Briot *et al*, 2015). Therefore, we speculated that Notch signaling in ECs could also be involved in the control of systemic glucose metabolism.

# Results

### Obesity induces endothelial Notch signaling

Notch signaling activity is altered by metabolites in several cell types (Pajvani *et al*, 2011, 2013; Bi *et al*, 2014; Briot *et al*, 2015). Therefore, we hypothesized that ECs, which are in intimate contact with plasma, respond to altered plasma metabolite concentrations. To examine this, we utilized diet-induced obesity (DIO) mouse models where C57BL/6J male mice were put on high-fat diet (HFD,

60% fat), high-fat-and-sucrose diet (HFS, 60% fat and 42 g/l sucrose in drinking water *ad libitum*), and a matching control diet (CD, 10% fat) for a period of 26 weeks starting at 4 weeks of age. We analyzed primary skeletal muscle ECs freshly isolated from these mice. Expression of Notch target genes was elevated in ECs isolated from obese animals (HFD and HFS) compared to ECs derived from CD fed mice (Fig 1A). In addition, we performed similar analyses with ECs isolated from skeletal muscle of mice on HFD for 3 and 8 weeks. Although these mice had a notable elevation in their blood glucose levels and body weights, the analysis of Notch target genes did not reveal any significant differences (Fig EV1A and B). Furthermore, to distinguish between a chronic and acute response to alterations in plasma metabolites, we examined Notch signaling during physiological feeding and fasting cycles. We analyzed Notch targets in primary ECs freshly isolated from skeletal muscle of mice that were either fed *ad libitum* or fasted for 24 h or refed for 6 h after a 24 h fast. We did not observe any significant differences in Notch target gene expression in ECs among these groups (Fig 1B). These results support the notion that chronic disturbance of plasma metabolites in obese mice leads to an increase in Notch signaling in ECs.

### Sustained endothelial Notch signaling lowers insulin sensitivity in mice

Notch signaling regulates glucose metabolism in liver and adipose tissue, and Notch over-activation in these tissues impairs insulin sensitivity (Pajvani *et al*, 2011; Bi *et al*, 2014). To test if sustained over-activation of Notch signaling, specifically in ECs, would also affect systemic glucose metabolism, we employed a mouse model in which constitutively active Notch1 intracellular domain (NICD) is expressed under the EC-specific tamoxifen-inducible *Cdh5* (VE-Cadherin) promoter (NICD[iOE-EC] mice; Ramasamy *et al*, 2014). Gene recombination was induced in adult mice (Wieland *et al*, 2017; Jabs *et al*, 2018). Seven weeks after gene recombination, there were no differences in body weight (Fig EV1C). Canonical Notch targets were significantly increased in ECs freshly isolated from skeletal muscle of NICD[iOE-EC] mice compared to controls (Fig EV1D). The range of *Hey1*, *Hey2*, and *Hes1* gene induction was very similar to the induction seen in diet-induced obese mice (Fig 1A). Furthermore, there were no differences in skeletal and cardiac muscle microvessel density and morphology in NICD[iOE-EC] mice when compared to littermate controls (Fig EV1E and F). However, upon EC Notch1 over-activation, the mice had increased plasma glucose and insulin levels (Fig 1C and D), a typical sign of impaired insulin sensitivity.

To confirm this observation, we performed an insulin tolerance test (ITT). Compared to controls, insulin lowered blood glucose less efficiently in NICD[iOE-EC] mice (Fig 1E and F). In addition, intraperitoneal glucose tolerance test (GTT) showed a similar trend (Fig 1G and H). To rule out impaired secretion of insulin or other hormonal regulators of glucose metabolism from pancreas and liver, we checked for vascular alterations in stained tissue sections. We did not observe any significant difference in pancreatic islet area or vessel coverage in pancreas between control and NICD[iOE-EC] mice (Fig EV2A and B). Moreover, histological analysis of liver vasculature also did not reveal any significant difference between the two groups (Fig EV2C and D), confirming that the observed insulin

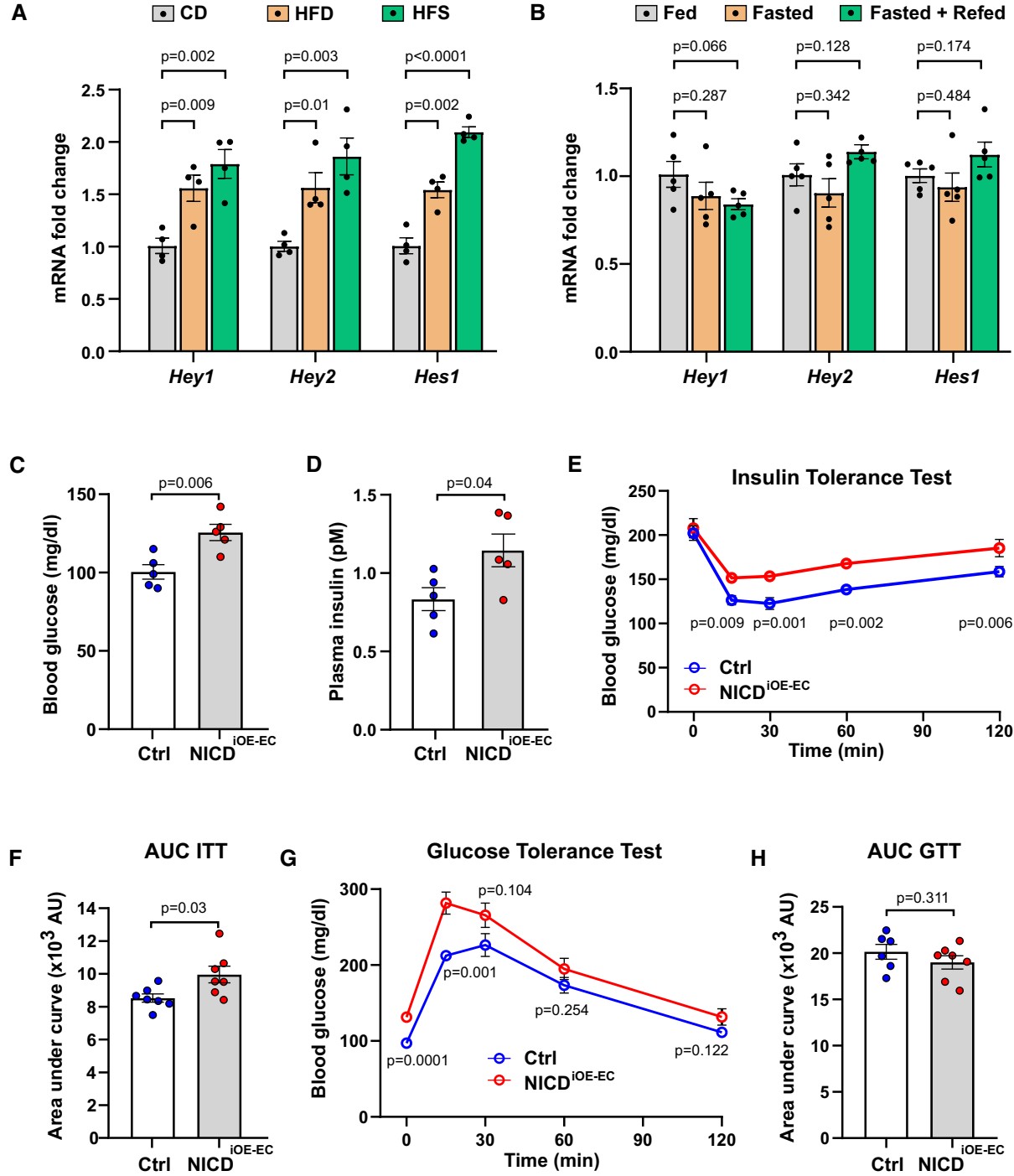

**Figure 1. Endothelial Notch signaling regulates systemic glucose metabolism.**

A   Expression of endothelial Notch target genes in microvascular endothelial cells isolated from skeletal muscle of mice kept on control diet (CD, 10% fat, 70% carbohydrates) or high-fat diet (HFD, 60% fat, 20% carbohydrates) or high-fat and sugar diet (HFS, 60% fat, 20% carbohydrates, and 42 g/l sucrose in drinking water). $n = 4$, data represent mean ± SEM, unpaired $t$-test.

B   Expression of endothelial Notch target genes in microvascular endothelial cells isolated from skeletal muscle of mice fasted for 24 h (Fasted group) and then refed for 4 h (Fasted + Refed group) normalized to mice fed for 24 h (Fed group). $n = 5$, data represent mean ± SEM, unpaired $t$-test.

C   Blood glucose levels of control ($n = 5$) and NICD[iOE-EC] ($n = 5$) mice 5 weeks after recombination. Data represent mean ± SEM, unpaired $t$-test.

D   Plasma insulin levels of control ($n = 5$) and NICD[iOE-EC] ($n = 5$) mice 5 weeks after recombination. Data represent mean ± SEM, Welch's $t$-test.

E   Blood glucose levels for insulin tolerance test of control ($n = 7$) and NICD[iOE-EC] control ($n = 7$) mice. Data represent mean ± SEM, unpaired $t$-test.

F   Quantification of area under curve for insulin tolerance test in (E). Data represent mean ± SEM, Welch's $t$-test.

G   Blood glucose levels for glucose tolerance test of control ($n = 6$) or NICD[iOE-EC] ($n = 7$) mice. Data represent mean ± SEM, unpaired $t$-test.

H   Quantification of area under the curve for glucose tolerance test in (G). Data represent mean ± SEM, unpaired $t$-test.

resistance in NICD[iOE-EC] mice is not an outcome of defective pancreas or liver function. Taken together, these data indicate that induced Notch signaling in ECs contributes to systemic insulin resistance.

## Inactivation of endothelial Notch signaling improves insulin sensitivity

To test if inhibition of EC Notch signaling would improve insulin sensitivity, we inactivated the *Rbpj* gene encoding Rbp-jκ, the essential transducer of signal transduction downstream of all four Notch receptors, specifically in ECs. Tamoxifen-driven genetic deletion of *Rbpj* in adult mice (*Rbpj*[iΔEC]) (Ramasamy *et al*, 2014; Jabs *et al*, 2018) did not affect body weight compared to littermate controls (Fig EV3A) but led to changes in vascular morphology and microvessel density in skeletal muscle tissue as we have previously described (Jabs *et al*, 2018). Five weeks after gene inactivation, *Rbpj*[iΔEC] mice had lower glucose and insulin levels in blood (Fig 2A and B). In addition, insulin-mediated lowering of blood glucose was

more pronounced and lasted longer in *Rbpj*[iΔEC] mice compared to controls (Fig 2C and D). Moreover, *Rbpj*[iΔEC] mice showed better tolerance to glucose (Fig 2E and F). Importantly, these metabolic alterations occurred before the onset of heart failure, which we had observed in our previous study around 7 weeks after gene inactivation (Jabs *et al*, 2018).

To exclude the possibility that the observed differences in systemic glucose homeostasis in *Rbpj*[iΔEC] mice are due to defects in pancreas and liver function, we performed extensive analysis on these tissues. We did not observe any significant difference in pancreatic islet area (Fig EV3B and C). The blood vessel area in pancreatic islet sections was increased (Fig EV3D), similar to what we had observed in skeletal and cardiac muscle (Jabs *et al*, 2018). Since the observed differences in vascularization of pancreatic islets could affect insulin secretion, we first performed an *ex vivo* glucose-stimulated insulin secretion (GSIS) assay from freshly isolated pancreatic islets. There was no difference in the total insulin content in islets isolated from *Rbpj*[iΔEC] mice compared to littermate controls (Fig EV3E). In addition, insulin secretion after adding 5.6 mM

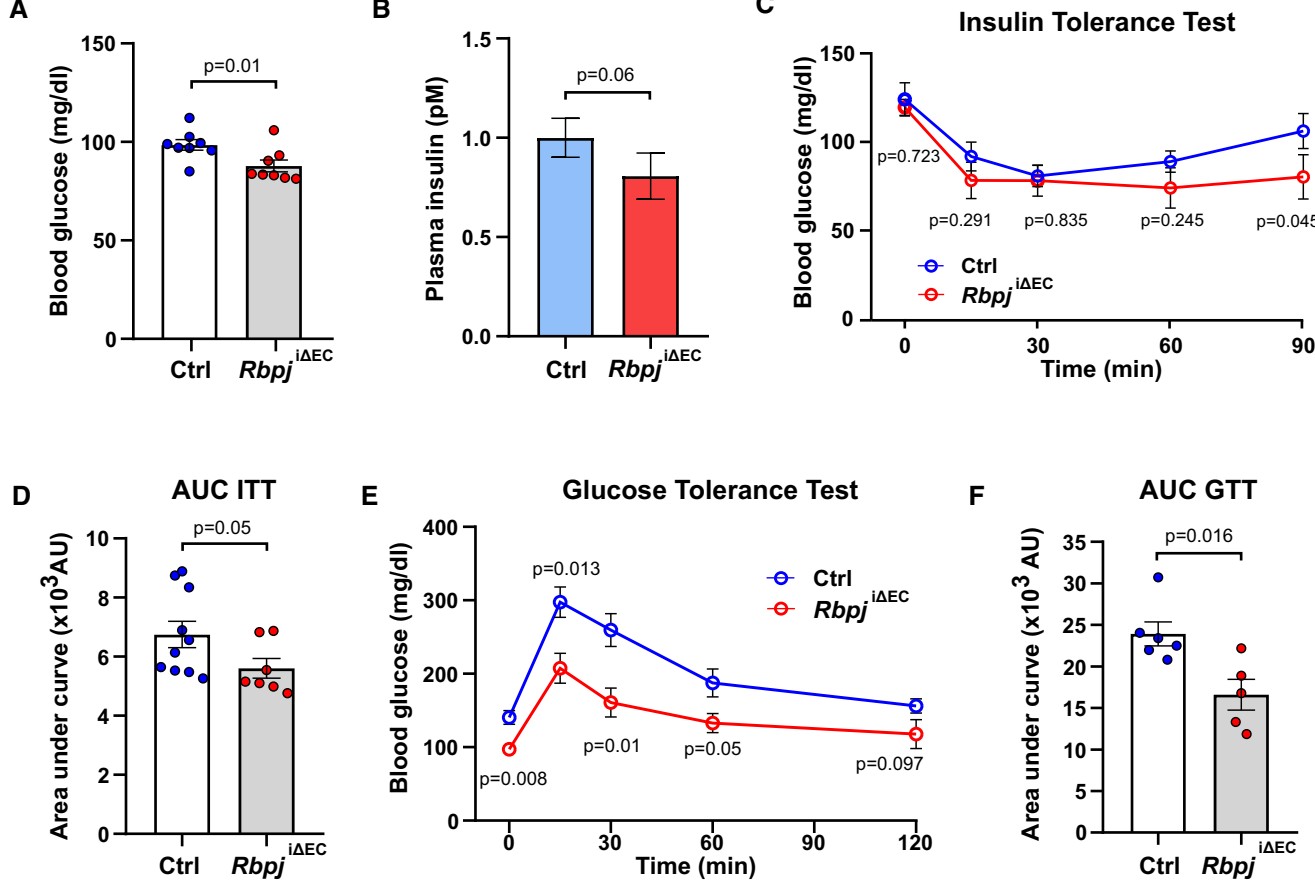

**Figure 2. Endothelial-specific Notch deletion improves insulin sensitivity.**

A   Blood glucose levels of control (*n* = 8) and *Rbpj*[iΔEC] (*n* = 8) mice 5 weeks after recombination. Data represent mean ± SEM, unpaired *t*-test.
B   Plasma insulin levels of control (*n* = 8) and *Rbpj*[iΔEC] (*n* = 8) mice 5 weeks after recombination. Data represent mean ± SEM, unpaired *t*-test.
C   Blood glucose levels for insulin tolerance test of control (*n* = 10) and *Rbpj*[iΔEC] (*n* = 7) mice. Data represent mean ± SEM, unpaired *t*-test.
D   Quantification of area under curve for insulin tolerance test in (C). Data represent mean ± SEM, Welch's *t*-test.
E   Blood glucose levels for glucose tolerance test of control (*n* = 6) or *Rbpj*[iΔEC] (*n* = 5) mice. Data represent mean ± SEM, unpaired *t*-test.
F   Quantification of area under the curve for glucose tolerance test in (E). Data represent mean ± SEM, Welch's *t*-test.

glucose was not altered. Only hyperglycemic conditions led to an increased insulin secretion in this simplified model (Fig EV3F). We subsequently measured plasma C-peptide levels after glucose administration *in vivo*, as insulin secretion is regulated by multiple factors. Importantly, this did not reveal any significant differences between control and $Rbpj^{i\Delta EC}$ mice (Fig EV3G and H).

Similar to pancreatic islet and muscle tissue, liver sections from $Rbpj^{i\Delta EC}$ mice also had higher microvessel density and vessel area (Fig EV4A and B). Despite the previously described pericentral sinusoidal dilation (Cuervo et al, 2016), we did not detect any hepatic inflammation, necrosis, fibrosis, or iron deposition in histological sections (Fig EV4C–E). In addition, albumin, urea, transaminases, and alkaline phosphatase levels were within the normal limits in plasma from both the groups (Fig EV4F–J).

Taken together, these results demonstrate that inhibition of canonical Notch signaling in ECs improves systemic insulin sensitivity.

## Notch signaling limits trans-endothelial insulin transport

Muscular ECs actively transport insulin via caveolae in continuous endothelial beds to muscle cells, which subsequently take up more glucose (Yang et al, 1989; Wang et al, 2006). Thereby, insulin is first taken up into ECs and later released at the basolateral membrane into the interstitial space. To test whether Notch signaling controls insulin uptake in ECs, we stimulated Notch signaling in cultured human umbilical venous ECs (HUVECs) with recombinant, immobilized Notch ligand DLL4. Activation of Notch signaling in these cells resulted in reduced uptake of FITC-labeled insulin (Fig 3A). Of note, the labeled insulin showed co-localization with CAV1, a marker protein for caveolae.

On the other hand, inhibition of canonical Notch signaling by expression of dominant-negative MAML1 (dnMAML) in HUVECs facilitated insulin internalization into ECs (Fig 3B).

Finally, we tested whether impaired EC Notch signaling would also enhance insulin transport *in vivo*. Insulin was injected intravenously into control and $Rbpj^{i\Delta EC}$ mice and the phosphorylation of protein kinase-B (AKT), a key mediator of insulin signaling, served as a readout for insulin transport. Phosphorylated AKT was measured in muscle lysates 7 min after injection. AKT phosphorylation was substantially higher in muscle tissue of $Rbpj^{i\Delta EC}$ mice compared to controls (Fig 3C and D). No such changes were observed in liver tissue of $Rbpj^{i\Delta EC}$ mice (Fig 3E and F). This may well reflect functional differences of liver and muscle ECs since the sinusoidal liver endothelium allows paracellular flux of molecules like insulin, whereas the continuous endothelium in muscle requires caveolae-mediated transcytosis for insulin transport (Aird, 2007). These data indicate that loss of endothelial Notch signaling *in vivo* increases trans-endothelial insulin transport to muscle cells.

## Caveolae facilitate insulin transport through cardiac endothelial cells

Vesicles in the continuous endothelium mediate transport of insulin from blood, across ECs to the interstitial space. Either clathrin-coated vesicles or caveolae have been implicated in this process based on tissue specificities (Wang et al, 2011; Azizi et al, 2015). To study trans-endothelial insulin flux, we employed a

transwell assay where a monolayer of mouse cardiac muscle ECs (MCECs) was cultured on the transwell and differentiated C2C12 myotubes were cultured in the lower chamber (Fig 4A). Different doses of insulin were added in the top chamber containing MCECs. The phosphorylation of AKT in C2C12 cells was monitored to assess insulin transport. AKT activity in C2C12 cells strongly correlated with the dose of insulin added to the MCECs (Fig 4B).

Caveolin proteins are the major structural proteins of caveolae and necessary for their formation (Sowa, 2012; Hansen et al, 2013). Several components of the caveolar system are expressed in ECs (Hansen et al, 2013). As we had previously detected co-localization of insulin with the caveolae marker protein Cav1 (Fig 3A and B) in ECs, we wanted to further investigate the role of Cav1 in mediating insulin transcytosis. *In vitro* silencing of Cav1 expression in MCECs reduced insulin transport rates to C2C12 myotubes in a transwell assay (Fig 4A, C and D), which is in agreement with previous reports (Wang et al, 2011).

A recent study has demonstrated that insulin receptors on ECs are needed for accelerated trans-endothelial insulin transport to muscle cells (Williams et al, 2018). By using a proximity ligation assay (PLA; Nordstrom et al, 2013), we observed that treatment of MCECs with insulin induced localization of Cav1 into close proximity with the insulin receptor. The close association between insulin receptor and Cav1 decreased 4 min after insulin treatment (Fig 4E), suggesting that insulin transiently sequesters its receptor to Cav1 during transcytosis.

## Endothelial Notch signaling inhibits the expression of proteins for caveolae formation

Next, we wanted to explore the link between Notch signaling and components of the caveolar system. To dissect this, we modulated Notch signaling in cultured HUVECs via adenoviral transfection of Notch1 ICD (NICD) or dnMAML and analyzed the expression of caveolar genes. Stimulation of Notch signaling by expression of NICD reduced *CAV1, CAV2,* and *CAVIN1* expression levels (Fig 5A), while the expression of *CAVIN2* and *CAVIN3* remained unchanged (Fig EV5A).

Consistent with these findings, the inhibition of Notch signaling by expression of dnMAML increased mRNA levels of *CAV1, CAV2,* and *CAVIN1* in HUVECs, while the expression of *CAVIN2* and *CAVIN3* was not changed (Figs 5A and EV5A). Adenoviral transfection of both dnMAML and NICD constructs in HUVECs was verified by evaluating expression levels of canonical Notch target genes. As expected, dnMAML overexpression led to downregulation of Notch targets and vice versa was observed with NICD overexpression (Fig EV5B). We also observed differences in CAV1 and CAVIN1 protein expression upon Notch manipulation (Fig 5B–E). However, we did not detect any significant differences in CAV2 protein levels (Fig EV5C and D).

Notch signaling typically inhibits gene expression via induction of the transcriptional repressors of the *Hey* and *Hes* gene family. It has been reported that Hey1 repressor physically binds the *Cav1* promoter and thereby downregulate its expression (Heisig et al, 2012). Similarly in our setting, transduction of HEY1 in HUVECs downregulated the expression of *CAV1*. Furthermore, expression of HEY1 reverted the induction of *CAV1* after Notch inhibition (Fig 5F)

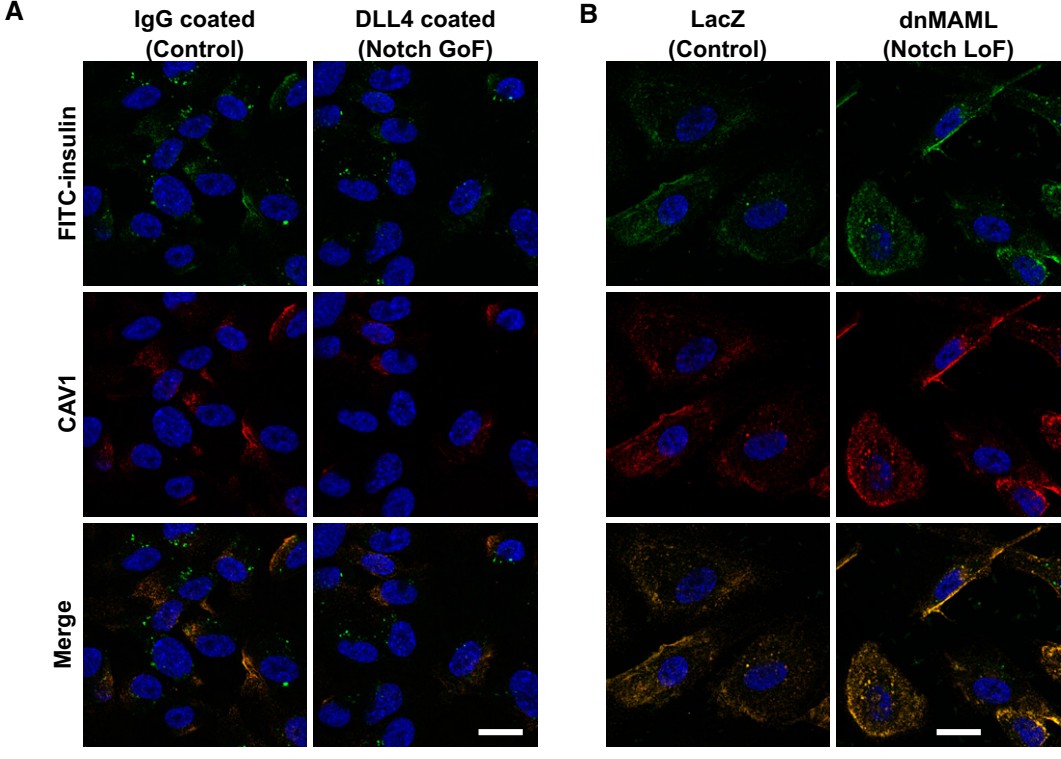

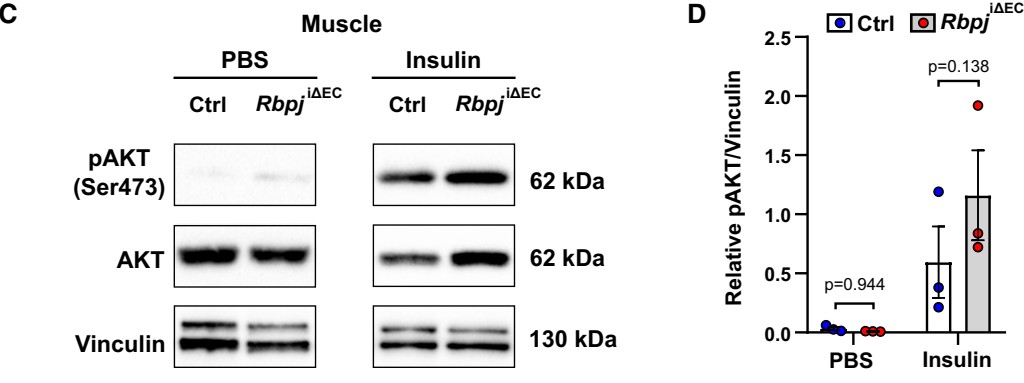

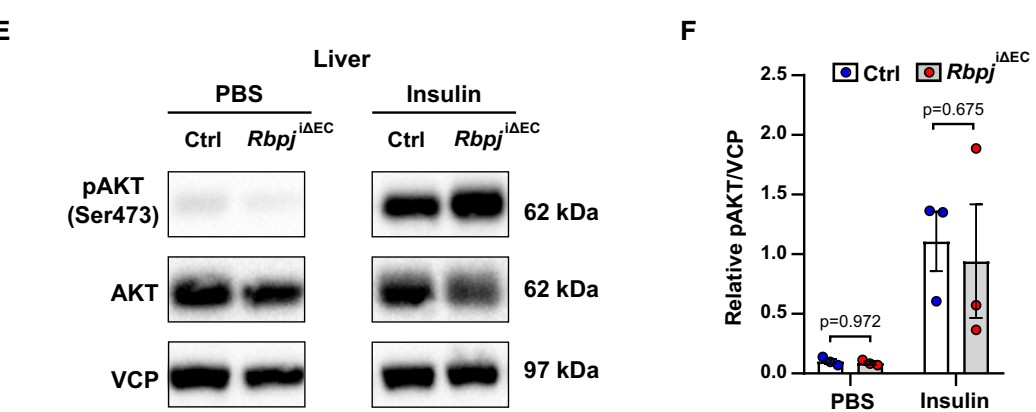

**Figure 3.**

**Figure 3.   Loss of Notch signaling improves endothelial insulin transport.**

A   Uptake of FITC insulin and CAV1 expression in primary human umbilical venous ECs (HUVECs) upon Notch induction. Scale bar 20 μm.
B   Uptake of FITC insulin and CAV1 expression in (HUVECs) upon Notch blockage. Scale bar 20 μm.
C   Representative Western blot of AKT phosphorylation in gastrocnemius muscle 7 min after intravenous injection of PBS or 1.5 U/kg insulin in control (n = 3) and
     *Rbpj*[iΔEC] (n = 3) mice 5 weeks after recombination.
D   Densitometric analysis of Western blot in (C). n = 3, data represent mean ± SEM, unpaired *t*-test.
E   Representative Western blot of AKT phosphorylation in liver 7 min after intravenous injection of PBS or 1.5 U/kg insulin in control (n = 3) and *Rbpj*[iΔEC] (n = 3) mice
     5 weeks after recombination.
F   Densitometric analysis of Western blot in (E). n = 3, data represent mean ± SEM, unpaired *t*-test.

Source data are available online for this figure.

in HUVECs, implying that Notch regulates *CAV1* expression through HEY1.

Next, we wanted to test whether the observed transcriptional changes in caveolar gene expression upon Notch modulation *in vitro* are also reproducible in murine skeletal muscle ECs. In primary ECs isolated from skeletal muscle of *Rbpj*[iΔEC] mice, mRNA levels of *Cav1, Cav2,* and *Cavin1* were elevated compared to controls (Fig 5G). Correspondingly, levels of *Cav1* and *Cavin1* were

downregulated in ECs freshly isolated from skeletal muscle of NICD[iOE-EC] mice (Fig 5H). However, we did not see any differences in *Cav2* mRNA levels (Fig 5H). Since we had previously detected upregulation of Notch targets in DIO mouse models, we decided to quantify the expression of the caveolar genes in these animals as well. Expression of *Cav1* and *Cavin1* was downregulated in ECs isolated from skeletal muscles of obese animals (HFD and HFS) compared to ECs derived from skeletal muscle of CD fed mice

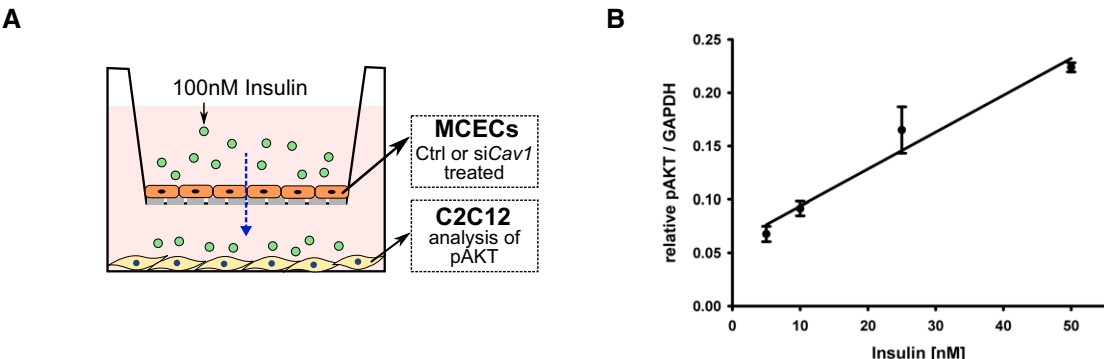

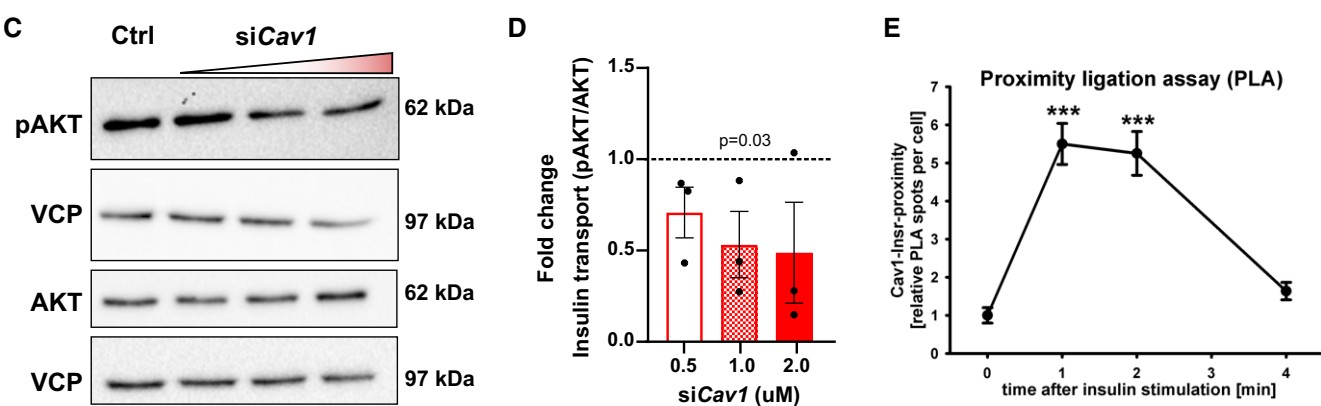

**Figure 4.   *Cav1*-dependent insulin transport across endothelial cells.**

A   Schematic representation of experimental setup for trans-endothelial insulin transport assay.
B   AKT phosphorylation in myogenic C2C12 cells treated with different doses of insulin. n = 3, data represent mean ± SEM, unpaired *t*-test.
C   Representative Western blot of AKT phosphorylation in C2C12 cells upon *Cav1* knockdown in mouse cardiac endothelial cells (MCECs).
D   Densitometric analysis of Western blot in (C). Data represented as fold change over control. n = 3, data represent mean ± SEM, unpaired *t*-test.
E   Proximity ligation assay showing the association of CAV1 and INSR (insulin receptor) in MCECs upon stimulation with insulin. n = 86–120 cells from 8 microscopic
     high power fields per time point, data represent mean ± SEM, unpaired *t*-test, *** means $P < 0.001$.

Source data are available online for this figure.

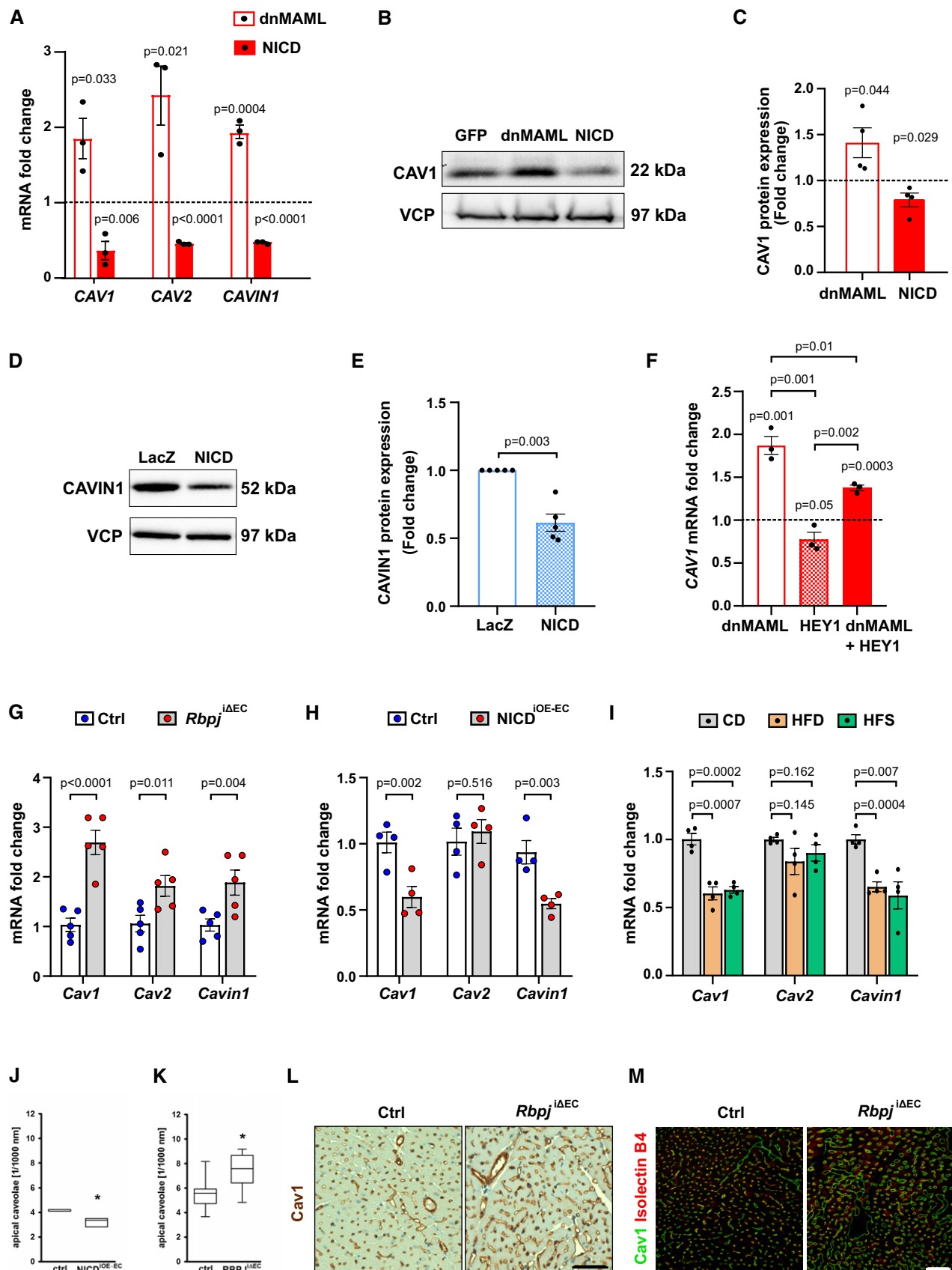

**Figure 5.**

**Figure 5.  Notch signaling controls insulin transport by expression of caveolar genes.**

A       Quantitative RT−PCR detection of *CAV1*, *CAV2*, and *CAVIN1* in primary human umbilical venous endothelial cells (HUVECs) upon Notch blockade (dnMAML) and induction (NICD). *n* = 3, data represent mean ± SEM, unpaired *t*-test.

B       Representative Western blot of CAV1 expression in HUVECs upon Notch manipulation.

C       Densitometric analysis of Western blot in (B). Data represented as fold change over GFP control. *n* = 3, data represent mean ± SEM, unpaired *t*-test.

D       Representative Western blot of CAVIN1 expression in HUVECs upon Notch induction.

E       Densitometric analysis of Western blot in (D). Data represented as fold change over LacZ control. *n* = 5, data represent mean ± SEM, unpaired *t*-test.

F       Expression of *CAV1* in HUVECs upon Notch manipulation and HEY1 induction. *n* = 3, data represent mean ± SEM, unpaired *t*-test.

G       Expression of *Cav1*, *Cav2*, *Cavin1* in primary microvascular ECs freshly isolated from skeletal muscle of *Rbpj*$^{\text{iΔEC}}$ mice compared to control mice. *n* = 5, data represent mean ± SEM, unpaired *t*-test.

H       Expression of *Cav1*, *Cav2*, *Cavin1* in primary microvascular ECs freshly isolated from skeletal muscle of NICD$^{\text{iOE-EC}}$ mice compared to control mice. *n* = 4, data represent mean ± SEM, unpaired *t*-test.

I       Expression of *Cav1*, *Cav2*, *Cavin1* in microvascular endothelial cells isolated from skeletal muscle of mice kept on control diet (CD, 10% fat, 70% carbohydrates) or high-fat diet (HFD, 60% fat, 20% carbohydrates) or high-fat and sugar diet (HFS, 60% fat, 20% carbohydrates, and 42 g/l sucrose in drinking water). *n* = 4, data represent mean ± SEM, unpaired *t*-test.

J, K    Quantification of number of caveolae attached to apical membrane in EC of cardiac capillaries from (J) NICD$^{\text{iOE-EC}}$ mice (6 segments of 1,000 nm length from 3 hearts per genotype) or (J) *Rbpj*$^{\text{iΔEC}}$ (6 segments of 1,000 nm length from 8 hearts per genotype) based on electron microscopy. Box plot depicts median and percentiles (10$^{\text{th}}$, 25$^{\text{th}}$, 75$^{\text{th}}$, 90$^{\text{th}}$), unpaired *t*-test, * means *P* < 0.05.

L, M    Representative immunohistochemistry (L) and confocal images (M) of Cav1 staining in heart sections from control and *Rbpj*$^{\text{iΔEC}}$ mice. *n* = 3, scale bar 50 μm.

Source data are available online for this figure.

(Fig 5I). Taken together, these data show that induction of Notch signaling decreases transcription of genes important for caveolae formation.

**Notch signaling limits the number of endothelial caveolae**

To test whether the decrease of caveolar proteins upon induction of EC Notch signaling would also lead to the formation of lower caveolae numbers, we performed electron microscopy of cardiac microvessels in NICD$^{\text{iOE-EC}}$ mice. Quantification of caveolae located at the luminal (apical) side of capillaries revealed that there were lower numbers of endothelial caveolae in NICD$^{\text{iOE-EC}}$ compared to littermate control animals (Fig 5J).

Consistently, we detected more caveolae at the luminal membrane in cardiac muscle ECs of *Rbpj*$^{\text{iΔEC}}$ mice (Fig 5K). Cav1 expression was also increased in *Rbpj*$^{\text{iΔEC}}$ cardiac muscle compared to controls (Fig 5L and M). These results substantiate the finding that EC Notch signaling drives a transcriptional program that is sufficient to change the number of endothelial caveolae in muscle tissue.

**Endothelial-specific Notch modulation alters glucose uptake in muscle**

Skeletal muscle tissue regulates systemic glucose homeostasis by serving as a major site for insulin-mediated glucose uptake. So far, our data show that stimulation of EC Notch signaling leads to changes in systemic glucose homeostasis characterized by higher glucose and plasma insulin levels, impaired insulin sensitivity, and reduced glucose clearance in NICD$^{\text{iOE-EC}}$ mice compared to controls. To determine muscle-specific contribution toward impaired systemic glucose homeostasis in these mice, we performed a glucose uptake assay where mice were injected with 2-deoxy glucose (2-DG) during a GTT. Blood samples were collected at different time points to measure glucose and plasma insulin levels. Finally, the mice were sacrificed 20 min after 2-DG injection to measure 2-DG uptake in different organs (Fig 6A). Blood glucose levels during the GTT followed a similar profile as observed before in Fig 1G, where

NICD$^{\text{iOE-EC}}$ mice showed reduced glucose clearance compared to controls (Fig 6B). Plasma insulin levels were also elevated in NICD$^{\text{iOE-EC}}$ animals (Fig 6C), indicating increased insulin resistance. Most importantly, analysis of 2-DG uptake showed lower glucose uptake in skeletal muscle of NICD$^{\text{iOE-EC}}$ mice (Fig 6D). We also measured 2-DG uptake in visceral white adipose tissue (vWAT), as adipose tissue is another site for insulin-mediated glucose uptake. However, we did not detect any significant difference in 2-DG uptake in vWAT between control and NICD$^{\text{iOE-EC}}$ mice (Fig 6E). These data suggest that higher plasma insulin along with reduced glucose uptake in muscle tissue in NICD$^{\text{iOE-EC}}$ animals leads to altered systemic glucose homeostasis.

We performed similar analysis in the EC-specific Notch loss-of-function *Rbpj*$^{\text{iΔEC}}$ mouse model (Fig 6F). Blood glucose levels during the GTT followed a similar profile as observed in Fig 2E, where *Rbpj*$^{\text{iΔEC}}$ mice showed improved glucose clearance compare to controls (Fig 6G). Plasma insulin levels were also reduced in *Rbpj*$^{\text{iΔEC}}$ animals (Fig 6H), indicating improved insulin sensitivity. Importantly, analysis of 2-DG uptake showed higher glucose uptake in skeletal muscle of *Rbpj*$^{\text{iΔEC}}$ mice (Fig 6I). We did not detect any significant difference in 2-DG uptake in vWAT between control and *Rbpj*$^{\text{iΔEC}}$ mice (Fig 6J).

In conclusion, these data underscore the role of endothelial-specific Notch signaling in maintaining systemic glucose homeostasis by altering glucose uptake in skeletal muscle.

**Endothelial-specific *Rbp-jκ* ablation improves glucose tolerance in a high-fat diet-induced insulin resistance model**

So far, our data show that inhibition of EC Notch signaling leads to increased number of caveolae, higher insulin transport rates, enhanced glucose uptake in skeletal muscle, and improved insulin sensitivity in *Rbpj*$^{\text{iΔEC}}$ mice compared to controls. Therefore, we wanted to investigate further whether targeting Notch signaling in the endothelium alone would be sufficient to lower blood glucose levels in a DIO mouse model that causes insulin resistance. To test this, mice were fed either a HFD (60% fat) or a matched CD (10% fat) for 8 weeks followed by gene recombination (Fig 7A). Analyses

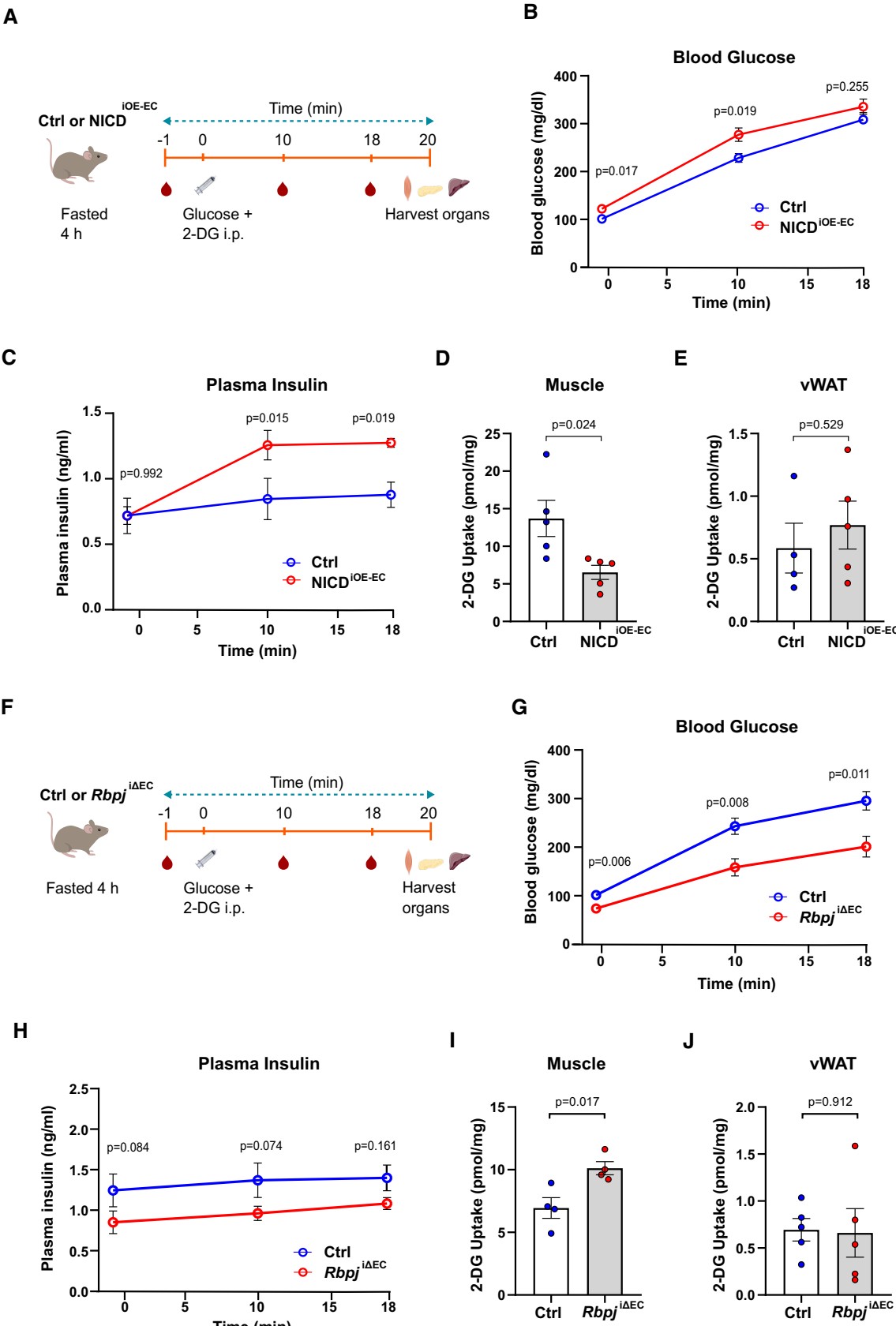

**Figure 6.**

**Figure 6. Endothelial-specific Notch modulation alters glucose uptake in muscle.**

A  Schematic illustration of 2-DG uptake assay protocol for control and NICD[iOE-EC] mice.
B  Blood glucose levels of control and NICD[iOE-EC] mice during the 2-DG uptake assay. $n = 5$, data represent mean $\pm$ SEM, unpaired $t$-test.
C  Plasma insulin levels of control and NICD[iOE-EC] mice during the 2-DG uptake assay. $n = 5$, data represent mean $\pm$ SEM, unpaired $t$-test.
D  2-DG uptake levels in skeletal muscle of control and NICD[iOE-EC] mice. $n = 5$, data represent mean $\pm$ SEM, unpaired $t$-test.
E  2-DG uptake levels in visceral white adipose tissue (vWAT) of control ($n = 4$) and NICD[iOE-EC] ($n = 5$) mice. Data represent mean $\pm$ SEM, unpaired $t$-test.
F  Schematic illustration of 2-DG uptake assay protocol for control and Rbpj[iΔEC] mice.
G  Blood glucose levels of control and Rbpj[iΔEC] mice during the 2-DG uptake assay. $n = 5$, data represent mean $\pm$ SEM, unpaired $t$-test.
H  Plasma insulin levels of control and Rbpj[iΔEC] mice during the 2-DG uptake assay. $n = 5$, data represent mean $\pm$ SEM, unpaired $t$-test.
I  2-DG uptake levels in skeletal muscle of control and Rbpj[iΔEC] mice. $n = 4$, data represent mean $\pm$ SEM, unpaired $t$-test.
J  2-DG uptake levels in vWAT of control and Rbpj[iΔEC] mice. $n = 5$, data represent mean $\pm$ SEM, unpaired $t$-test.

were done between 5 and 7 weeks postrecombination while maintaining the diet plan. There were no changes in body weight between Rbpj[iΔEC] and control mice in both diet groups (Fig EV5E and F). However, in both diet groups, Rbpj[iΔEC] mice had significantly lower blood glucose levels compared to controls starting 5 weeks after gene inactivation (Fig EV5G and H).

In addition, GTT of HFD fed control and Rbpj[iΔEC] mice showed significantly enhanced glucose tolerance in Rbpj[iΔEC] mice compared to control mice (Fig 7B and C). Similarly, ITT of HFD fed control and Rbpj[iΔEC] mice showed remarkable improvement in insulin sensitivity in Rbpj[iΔEC] mice compared to controls (Fig 7D and E). We analyzed glycated hemoglobin (HbA1c), a long-term marker of average blood glucose levels, in both groups on HFD. Consistently, HbA1c levels were lower in Rbpj[iΔEC] mice compared to controls (Fig 7F).

Lastly, we also measured 2-DG uptake in control and Rbpj[iΔEC] mice kept on HFD (Fig 7G). Blood glucose levels during the GTT followed a similar profile as observed in Fig 7B, where Rbpj[iΔEC] mice showed improved glucose clearance compare to controls (Fig 7H). Plasma insulin levels were also reduced in Rbpj[iΔEC] animals (Fig 7I), indicating enhanced insulin sensitivity. Furthermore, analysis of 2-DG uptake showed higher glucose uptake in skeletal muscle of Rbpj[iΔEC] mice on HFD (Fig 7J). Interestingly, we also detected significant increase in 2-DG uptake in vWAT in Rbpj[iΔEC] mice on HFD mice (Fig 7K). In conclusion, these results indicate that endothelial-specific Notch inhibition improves glucose homeostasis in a HFD induced insulin resistance model.

## Discussion

This study demonstrates an eminent function of the endothelium for the control of systemic glucose homeostasis. We could show that ECs not only react to changes in plasma metabolite concentration in obese mice, but also actively respond by altering flux of insulin across the continuous endothelium to muscle cells.

Diet-induced obesity (DIO) mouse models exhibit numerous systemic metabolic changes like hyperlipidemia, hyperglycemia, and hyperinsulinemia (Kim et al, 2004). The observed increase in Notch signaling in the endothelium of obese mice kept for long term on high-fat diet could be due to single or multiple factors. Several recent studies have reported hyperglycemia induced upregulation of Notch signaling both in the endothelium and in other cell types (Pajvani et al, 2011, 2013), which supports our finding.

Recent work from our laboratory has implicated that endothelial-specific Notch signaling inhibition by genetic ablation of Rbpj impairs long-chain fatty acid transport to cardiomyocytes (Jabs et al, 2018). However, in this study, we show in the same mouse model

that endothelial-specific Notch inhibition improves glucose tolerance and insulin sensitivity. These results at a glance might seem conflicting in terms of defining the physiological role of Notch signaling in regulating metabolism. Our data show that physiological changes during feeding cycles do not strongly affect Notch signaling in ECs. However, chronic metabolic disturbances such as those seen in obesity might lead to sustained over-activation of Notch signaling activity in ECs and this subsequently would contribute to impaired insulin sensitivity. As such, it is important to note that the amplitude of Notch activation could differ between physiological and pathological conditions (Kopan, 2012), and that in particular, chronic over-activation affects systemic glucose metabolism.

Notch signaling is a master regulator of blood vessel development and homeostasis. Perturbations in Notch signaling often lead to alterations in vascular morphology during development (Tetzlaff & Fischer, 2018). We had shown before that the increase in microvessel density does not significantly alter tissue perfusion in the Rbpj[iΔEC] mice (Jabs et al, 2018). In this study, we had observed increased microvessel density in pancreatic islets and liver. However, this did not have a profound effect on organ function.

In addition, a recent study has demonstrated that non-canonical Notch signaling limits vascular permeability (Polacheck et al, 2017). In previous studies from our laboratory, we had observed increased permeability in NICD-expressing HUVEC monolayer (Wieland et al, 2017), and Notch inhibition increased paracellular permeability as determined by trans-endothelial electrical resistance and capacity measurements in HUVECs (Jabs et al, 2018). Therefore, we did not include any in vitro experiments to measure insulin flux across endothelial monolayer upon Notch manipulation in this study. However, in vivo, as described previously (Jabs et al, 2018), we did not see any increase in extravasation of Evans blue between Rbpj[iΔEC] mice and corresponding littermate controls. Hence, the increased insulin flux across the endothelium observed in Rbpj[iΔEC] mice is most likely not due to grossly altered paracellular permeability.

Mechanistically, the experiments revealed that endothelial Notch signaling limits caveolae formation. Although trans-endothelial insulin transport varies among different organs, caveolae-mediated insulin flux is one of the well-described mechanisms (Wang et al, 2011; Azizi et al, 2015). Our experiments indicated that Notch signaling in ECs leads to transcriptional repression of Cav1 and Cavin1 expression, which are both essential components for caveolae formation. Notch signaling inhibits gene expression via induction of the transcriptional repressors of the Hey and Hes gene families. It has been shown that Hey1 physically binds to the Cav1 promoter (Heisig et al, 2012). Several publications have shown that loss of expression of one caveolar protein causes a downregulation of other proteins involved in caveolae formation (Davalos et al,

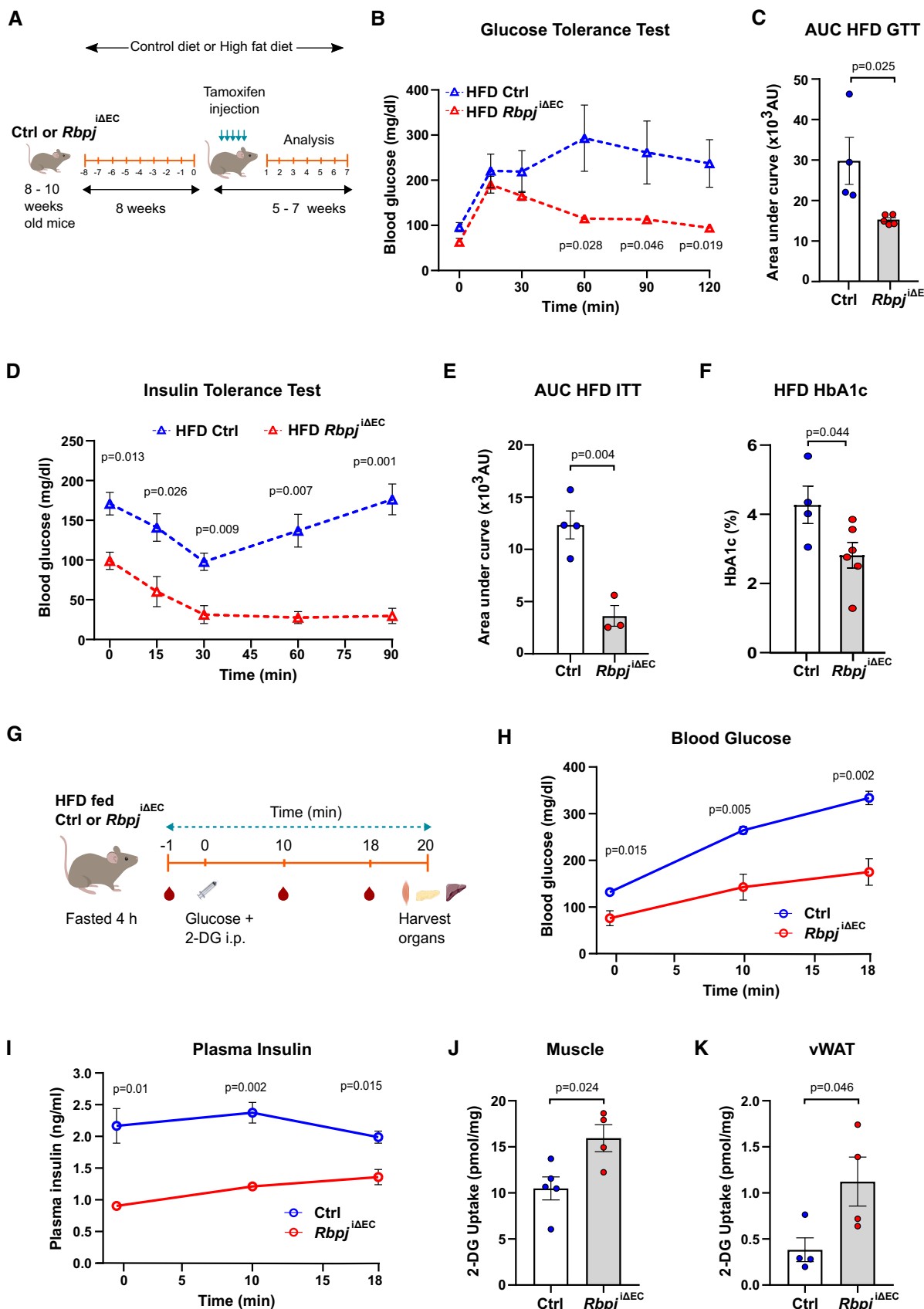

**Figure 7.**

**Figure 7. Endothelium-specific deletion of *Rbpj* improves glucose tolerance in obese mice.**

- A Schematic illustration of feeding and recombination protocol.
- B Blood glucose levels for glucose tolerance test (GTT) of control (*n* = 4) or *Rbpj*[iΔEC] (*n* = 5) mice kept on high-fat diet (HFD). Data represent unpaired *t*-test, mean ± SEM.
- C Quantification of area under curve (AUC) for GTT in (B). Data represent mean ± SEM, unpaired *t*-test.
- D Blood glucose levels for insulin tolerance test (ITT) of control (*n* = 4) or *Rbpj*[iΔEC] (*n* = 3) mice kept on HFD. Data represent unpaired *t*-test, mean ± SEM.
- E Quantification of AUC for ITT in (D). Data represent mean ± SEM, unpaired *t*-test.
- F HbA1c levels of control (*n* = 4) or *Rbpj*[iΔEC] (*n* = 6) mice kept on HFD 7 weeks after recombination. Data represent mean ± SEM, unpaired *t*-test.
- G Schematic illustration of 2-DG uptake assay protocol for control and *Rbpj*[iΔEC] mice on HFD.
- H Blood glucose levels of control and *Rbpj*[iΔEC] mice on HFD during 2-DG uptake assay. *n* = 5, data represent mean ± SEM, unpaired *t*-test.
- I Plasma insulin levels of control and *Rbpj*[iΔEC] mice on HFD during 2-DG uptake assay. *n* = 5, data represent mean ± SEM, unpaired *t*-test.
- J 2-DG uptake levels in skeletal muscle of control (*n* = 5) and *Rbpj*[iΔEC] (*n* = 4) mice on HFD. Data represent mean ± SEM, unpaired *t*-test.
- K 2-DG uptake levels in vWAT of control and *Rbpj*[iΔEC] mice. *n* = 4, data represent mean ± SEM, unpaired *t*-test.

2010; Hansen *et al*, 2013). Furthermore, we could show downregulation of *Cav1* and *Cavin1* in DIO mouse models compared to lean mice. These data are further supported by a recent publication where authors reported fewer endothelial vesicles of skeletal tissue in chronically obese male mice resulting in reduced trans-endothelial insulin transport (Williams *et al*, 2020).

Skeletal muscle is the predominant site for insulin-mediated glucose uptake, thereby regulating systemic glucose homeostasis. Insulin should cross the continuous endothelium in the muscle tissue to reach the myocytes. Sustained endothelial Notch signaling in NICD[iOE-EC] mice resulted in reduced glucose uptake in skeletal muscle, ultimately leading to impaired systemic glucose homeostasis. On the contrary, endothelial-specific Notch inhibition in *Rbpj*[iΔEC] mice leads to enhanced glucose uptake in skeletal muscle, thereby improving the insulin sensitivity in these mice. However, we did not see any difference in glucose uptake in vWAT, another insulin responsive tissue with continuous endothelium. These differences could be attributed to tissue-specific differences in endothelial properties between muscles and adipose tissue or due to differences in mode of insulin transport as shown by a recent publication that insulin transcytosis in adipose tissue ECs is clathrin dependent and caveolae independent (Azizi *et al*, 2015).

Taken together, this study reveals that the endothelium contributes to the onset and prevention of insulin resistance. Under physiological conditions, a certain degree of insulin resistance is important to prevent excessive glucose consumption by muscle cells, as glucose is essential to nourish neurons and erythrocytes. In diabetes mellitus, the degree of insulin resistance is often increased. Our study shows that using a genetic approach, endothelial Notch inhibition lowers blood glucose levels and substantially improves glucose tolerance in mice fed with a high-fat diet. Therefore, it is tempting to speculate that pharmacological targeting of endothelial Notch signaling could be beneficial. However, our previous study has demonstrated that chronic Notch inhibition in ECs impairs heart function. It would therefore be interesting testing short-term treatment regimens.

## Materials and Methods

### Animal models

The study was approved by Institutional and Regional Animal Research Committees. All animal procedures were in accordance with institutional guidelines and performed according to the guidelines of the local institution and the local government. Animals were group-housed under specific pathogen-free conditions. *Rbpj*[iΔEC] mice (Cdh5-CreERT2, Rbpj[lox/lox]) were obtained by crossing Cdh5-CreERT2 mice with *Rbpj*[lox/lox] mice. Mice overexpressing the Notch1-ICD in the endothelium (NICD[iOE-EC] mice) were obtained by crossing Gt(ROSA)26Sortm1(Notch1)Dam/J mice to Cdh5-CreERT2 mice (Ramasamy *et al*, 2014). *Rbpj*[iΔEC] mice, NICD[iOE-EC] mice, and floxed controls without CreERT2 alleles were injected 1.5 mg tamoxifen i.p. for five consecutive days. Gene recombination was induced at an age of 8–12 weeks. Male mice were used for experiments. Experiments were performed before the onset of heart failure. Extensive phenotyping of these mice had been reported before (Wieland *et al*, 2017; Jabs *et al*, 2018). Mice were generally kept on chow diet. When indicated, mice were fed a high-fat diet containing 60% fat, 20% carbohydrate, 20% proteins (D12492i, Research Diets) or a matching control diet with 10% fat, 70% carbohydrate, 20% proteins (D12450Bi, Research Diets). For the cohort with mice on high-fat and sugar diet, mice were fed a high-fat diet containing 60% fat, 20% carbohydrate, 20% proteins (D12492i, Research Diets) with 42 g/l sucrose in drinking water *ad libitum* or a matching control diet with 10% fat, 70% carbohydrate, 20% proteins (D12450Ji, Research Diets).

### Skeletal muscle microvascular EC isolation

For isolation of skeletal muscle microvascular ECs, gastrocnemius, soleus, and quadriceps were dissected and minced. Minced tissue was digested with 2 mg/ml collagenase II (PAN-Biotech) and 2.4 mg/ml dispase II (Sigma) in PBS for 1 h at 37°C with vortexing at 10-min interval. Digestion was quenched with PBS/1% BSA and the dissociated cells were filtered through 70-μm filter and centrifuged at 300 × *g* for 5 min. Supernatant was discarded and the cell pellet was resuspended in 3 ml PBS/1% BSA. Cell suspension was incubated for 1 h with Sheep anti-rat Dynabeads (Invitrogen) coupled to rat anti-mouse CD31-antibody (BD Pharmingen, #550274). Cells were washed three times with PBS and then lysed in RNA extraction buffer (PicoPure RNA Isolation Kit, Applied Biosystems). Beads were separated from the cell lysate and RNA was isolated according to the manufacturer's protocol.

### Quantitative PCR

RNA was isolated using the innuPREP RNA Mini kit (Analytik Jena). cDNA was synthesized with the High-Capacity cDNA Reverse Transcription Kit (Applied Biosystems). The cDNA was applied to qPCR

using the POWER SYBR Green Master Mix (Applied Biosystems). Fold changes were assessed by $2^{-\Delta\Delta Ct}$ method and normalized with the genes *OAZ1* (Human) and *Rpl13a* (mouse). The following primers were used for qPCR:

| Gene | Forward | Reverse |
| --- | --- | --- |
| *Hey1* | GCTCACCCAGACTACAGCTC | CAAGTTTCCATTCTCGTCCGC |
| *Hey2* | GGGTAAAGGCTACTTTGATGCC | ACTTCTGTCAAGCACTCTCGG |
| *Hes1* | AAAAATTCCTCCTCCCCGGT | GATAGGCTTTGATGACTTTCTGTGC |
| *Cav1* | ACGACGACGTGGTCAAGATT | GTGCAGGAAGGAGAGAATGG |
| *Cav2* | GCGTTGACTACGCAGATCCT | GCCAGAAATACGGTCAGGAA |
| *Cavin1* | CGCCGCAACTTCAAAGTCAT | CAGTGCCTCCGACTCTTTCA |
| *HEY1* | GAGAAGGCTGGTACCCAGTG | CGAAATCCCAAACTCCGATA |
| *HEY2* | CTTGTGCCAACTGCTTTTGA | GCACTCTCGGAATCCTATGC |
| *HES1* | TCAACACGACACCGGATAAA | CCGCGAGCTATCTTTCTTCA |
| *CAV1* | GAGCTGAGCGAGAAGCAAGT | CAAATGCCGTCAAAACTGTG |
| *CAV2* | AGTTCCTGACGGTGTTCCTG | CGTCCTACGCTCGTACACAA |
| *CAVIN1* | ACGAGCAATACGGTGAGCAA | CCTCCGACTCTTTCAGCGAT |
| *CAVIN2* | TTGTTTTCATCCGGAAGCTC | ACGCAACCATTTCAAAGTGC |
| *CAVIN3* | TCCGAGCTCTCTCCAACTTC | GAAGCTCCACGTTCTGCTCT |
| *OAZ1* | GAGCCGACCATGTCTTCATT | CTCCTCCTCTCCCGAAGACT |

## Cell culture

HUVECs were isolated from umbilical cords and maintained until passage 3 in endopan-3 growth medium containing 3% FCS and supplements (PAN-Biotech). MCECs were procured from Biozol/CELLutions Biosystems Inc (Catalogue No. CLU510). MCECs were grown on gelatin-coated surfaces in DMEM with 5% FCS, 1 g/l glucose, 10 mM Hepes. All cell lines were routinely tested for mycoplasma contamination.

HUVECs were transduced with adenoviral vectors at a MOI of 50. The vectors encoding dominant-negative MAML1, Notch1 intracellular domain, GFP and lacZ were described (Brutsch *et al*, 2010; Woltje *et al*, 2015). For experiments with fluorescence as readout, the mCherry cassette was removed using BamHI and NotI restriction enzymes.

## Immunostaining and FITC insulin uptake

HUVECs were transduced with adenoviral vectors and seeded the next day on coverslips coated with gelatin. Alternatively, non-transduced cells were seeded on coverslips coated with human IgG or human recombinant DLL4. The next day, cells were serum-starved for 4 h and were then incubated with basal medium containing 100 nM FITC insulin (Sigma) for 5 min. Cells were washed with PBS and fixed with 4% PFA for 20 min. Then, the coverslips were washed three times for 5 min with PBS and blocked for 1 h in 5% BSA in TBST. The coverslips were incubated with antibodies against CAV1 (Cell Signaling) overnight at 4°C. The coverslips were rinsed three times in TBST and were incubated with a secondary antibody coupled to Alexa Fluor-647 for 1 h. The coverslips were washed again and incubated with a DAPI solution before they were washed again. The coverslips were mounted and imaged with a confocal microscope.

## Insulin transport through an endothelial cell monolayer

MCECs were seeded at high density on cell culture inserts (pore size 0.4 μm) coated with fibronectin. The next day, cells were transfected with control or *Cav1* siRNA. Two days later, the cells were starved for 4 h and then the upper compartments were filled with medium containing 100 nM human insulin and the inserts were transferred to cell culture wells containing confluent C2C12 myotubes which had been serum-starved overnight. C2C12 myoblasts were differentiated in advance in high glucose DMEM containing 2% horse serum. After 10 min of co-culture, in which the insulin from the upper compartment could pass the endothelial monolayer and stimulate the cells in the lower compartment, the inserts were removed and the cells in the lower compartment were lysed. The amount of AKT phosphorylated at Ser-473 was determined by Western blot. As control for linearity of AKT phosphorylation, MCECs were starved overnight and were incubated for 20 min with 5, 10, 25, or 50 nM insulin before they were lysed.

## Western blotting

For protein analysis, cells or tissues were lysed in cell lysis buffer (#9803, Cell Signaling) containing 1 mM PMSF. Proteins were electrophoresed on 10% SDS gels and were transferred to nitrocellulose membranes, which were then blocked in 5% skim milk in TBS containing 0.05% Tween-20. The membranes were incubated with primary antibodies overnight at 4°C, washed, and incubated for 1 h with peroxidase-conjugated secondary antibody (Dako) at room temperature. Images were acquired with a ChemiDoc imaging system and quantified with Image Lab software (both Bio-Rad). The following primary antibodies were used CAV1 (1:1,000, ab2910), p-AKT Ser473 (1:1,000, Cell Sig. 4060), AKT (1:1,000, Cell Sig. 9272), VCP (1:1,000, ab11433), CAV2 (1:1,000, ab133484), CAVIN1 (1:1,000, ab48824), Vinculin (1:1,000, Sigma V9131).

## Proximity ligation assay

MCECs were seeded at low density onto coverslips and serum-starved overnight. Then, they were incubated with 100 nM insulin for the indicated period, washed, and fixed with 4% PFA before they were blocked for 1 h in 5% BSA in TBST. PLA was performed with primary antibodies against Cav1 (abcam ab2910) and Insr beta (Santa Cruz sc-31367). PLA was performed according to the manufacturer's guidelines (Duolink Orange Detection System, Olink Biosciences). PLA spots were counted with a fluorescence widefield microscope.

## Immunofluorescence and tissue histology

Stainings were performed on formalin-fixed paraffin-embedded sections (3 μm). Sections were deparaffinized and rehydrated. For hematoxylin and eosin, Prussian blue, and Sirius red staining, sections were processed according to standard protocols. For immunofluorescence stainings, antigen retrieval was performed followed by blocking and primary antibody incubation at 4°C overnight. The following primary antibodies were used: rabbit anti-mouse collagen IV (1:200, #2150-1470, Bio-Rad), guinea pig

anti-mouse insulin (1:50, ab7842, Abcam), isolectin-B4 (1:100, Life Technologies, #132450) with antigen retrieval at pH 6 with citrate buffer, rabbit anti-mouse CD31 (1:50, ab28364, Abcam) with antigen retrieval at pH 9 with citrate buffer. After washing, sections were incubated with secondary antibody (1:200) for 1 h at room temperature. The following secondary antibodies were used: goat anti-rabbit Alexa Fluor 647 (Life Technologies, A21245), goat anti-rabbit Alexa Fluor 546 (Life Technologies, A11035), and goat anti-guinea pig Alexa Fluor 546 (Life Technologies, A11074).

## Metabolic tests

Glucose was measured in blood taken from the tail vein with a Free-Style Lite blood sugar measuring system. Insulin was measured with a kit from DRG Diagnostics. Glycated hemoglobin (HbA1c) was determined by HPLC in whole-blood lysates using a PolyCAT A column (PolyLC). Hemoglobin was detected at 415 nm, and the ratio of glycated hemoglobin was assessed by comparing the peak area to that of unglycated hemoglobin.

For testing glucose tolerance, mice were fasted for 6 h before they received 2 g/kg body weight glucose by i.p. injection. Blood glucose was measured, and serum was sampled at indicated time points. For testing insulin tolerance, mice were starved for 4 h before 0.75 U human insulin per kg body weight was injected. To assess insulin accessibility to muscle and liver, mice were starved overnight before they received 1.5 U insulin per kg body weight i.v. After 7 min, the mice were sacrificed and organs were snap-frozen, lysed, and analyzed by Western blot detecting p-AKT. Total body weight was measured with laboratory scales.

For measurement of ASAT, ALAT, ALP, urea, and albumin, serum samples were analyzed in the Central Laboratory of Heidelberg University Hospital on a Siemens ADVIA® Chemistry XPT System according to the manufacturer's protocol.

## 2-DG uptake assay

For 2-DG uptake assay, mice were fasted for 4 h. 2-DG (300 μmol/kg) was injected i.p. along with glucose (2 g/kg body weight for fasted lean mice; 1 g/kg body weight for fasted HFD mice). Blood samples were collected from tail vein at baseline and after injection at different time points. Blood glucose was measured using Free-Style Lite blood sugar measuring system. Epididymal WAT (25–30 mg) and soleus muscle (10–15 mg) samples were collected 20 min after 2-DG injection and snap-frozen. Tissue samples were homogenized in 500 μl 10 mM Tris–HCl (pH = 8), heated at 95°C for 15 min, and centrifuged at $16,000 \times g$ at 4°C for 15 min. Supernatant was collected and used to measure 2-DG uptake using a 2-DG Uptake Measurement Kit (CSR-OKP-PMG-K01TE, Cosmo Bio). Plasma insulin levels were measured using Ultrasensitive Insulin ELISA Kit (90080, Crystal Chem).

## Glucose stimulated insulin secretion

For *ex vivo* GSIS, mouse islets were isolated by Collagenase P (Roche, Germany # 11213865001) digestion of the pancreas (Deglasse *et al*, 2019). After digestion, islets were washed, hand-picked under a stereomicroscope, and cultured overnight at 37°C

### The paper explained

#### Problem
Metabolic diseases such as diabetes mellitus frequently lead to subsequent blood vessel damage, impairing endothelial cell function. These damages pose a major risk for further metabolic complications and cardiovascular events. Despite their contribution to disease prevalence, the role of endothelial cells in actively regulating systemic metabolism is poorly understood.

#### Results
Here, we investigated how endothelial cells are involved in the control of systemic glucose metabolism. Our experiments revealed that Notch signaling in endothelial cells controls gene expression of proteins required for caveolae formation, which are in turn essential for insulin transport to muscle cells. Chronic over-activation of Notch signaling impaired insulin sensitivity and increased blood glucose levels. On the contrary, inhibition of Notch signaling increased insulin sensitivity and improved glucose tolerance.

#### Impact
Our data imply that the endothelium actively contributes to the control of insulin sensitivity and glucose uptake in muscle. Therefore, blood vessels not only transport insulin and glucose but also regulate their transport across the vessel wall.

and 5% $CO_2$ in RPMI medium 1640 containing 11 mmol/l glucose, 10% FBS, 100 U/ml penicillin, and 100 μg/ml streptomycin.

After overnight culture, batches of 10 islets were incubated for 1 h in a bicarbonate-buffered Krebs (KH) solution containing in (mmol/l): NaCl (120), KCl (4.8), CaCl₂ (2.5), MgCl₂ (1.2), NaHCO₃ (24), 1 g/l BSA, and glucose (0.5). Supernatant was discarded and islets were then incubated in Krebs buffer for 1 h in the presence of 5.6 mM or 16.7 mM glucose. Supernatant was collected and total insulin content was extracted with ethanol acid buffer (70% ethanol, 1.5% hydrochloric acid). Insulin release and total insulin content were measured by fluorescence resonance energy transfer (FRET) using the Insulin Ultrasensitive Assay kit (Cisbio, Codolet, France # 62IN2PEG). The signal intensity was measured at 665 and 620 nm using CLARIOstar Microplate Reader (BMG LABTECH).

For *in vivo* GSIS, mice were fasted for 16 h. Glucose (2 g/kg body weight) was injected i.p. Blood samples were collected from tail vein at baseline and at different time points after injection. Plasma C-peptide levels were measured using Ultrasensitive C-peptide ELISA Kit (90050, Crystal Chem).

## Electron microscopy

Hearts were fixed in Karnovsky's solution, followed by postfixation in 2% osmium tetroxide, and were then embedded in Araldite (Polysciences). Ultrathin sections were cut 60–70 nm with a Leica Ultracut UCT (Leica Microsystems), were counterstained with uranyl acetate and lead citrate, and were analyzed with an EM900 (Zeiss).

## Statistical analysis

GraphPad Prism 8 was used to generate graphs and for statistical analysis. Column statistics was performed on data to check for normality. Unpaired *t*-test or *t*-test with Welch's correction was used

accordingly. Data sets are presented as mean $\pm$ SEM. $P < 0.05$ was considered as significant.

**Expanded View** for this article is available online.

## Acknowledgements
We thank the Center for Preclinical Research and the Imaging Unit at the German Cancer Research Center for support. This work was supported in part by grants from the German Research Foundation DFG SFB-1366 (project number 394046768) to projects C4 and B3 to A.F. and C.G., respectively; DFG grant FI 1569/3-1 to A.F.; SFB1118 to T.F., S.H., H.J.G., and P.P.N.; SFB1123 to S.H.; SFB894 to C.C. and the Chica and Heinz Schaller foundation to A.F.

## Author contributions
SSH, MJ, JT, LW, TL, VN, GF, LPR, CC, GW, BE-Ü, TF, MB, IM, FT performed experiments and analyzed data; H-JG, TF, CG, PPN, and SH contributed to the planning and performance of the experiments and analysis of the data; SSH, MJ, and AF wrote the paper; and AF conceived and directed the study.

## Conflict of interest
The authors declare that they have no conflict of interest.

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
