## [Review Process File · EMBO Molecular Medicine]

Endothelial Notch signaling controls insulin transport in muscle

Sana S. Hasan, Markus Jabs, Jacqueline Taylor, Lena Wiedmann, Thomas Leibing, Viola Nordström, Giuseppina Federico, Leticia P. Roma, Christopher Carlein, Gretchen Wolff, Bilgen Ekim-Üstünel, Maik Brune, Iris Moll, Fabian Tetzlaff, Hermann-Josef Gröne, Thomas Fleming, Cyrill Géraud, Stephan Herzig, Peter P. Nawroth, Andreas Fischer

Review timeline:	Submission date:	26th Apr 2018
	Editorial Decision:	14th Jun 2018
	Revision received:	31st May 2019
	Editorial Decision:	4th Jul 2019
	Revision received:	3rd Feb 2020
	Editorial Decision:	18th Feb 2020
	Revision received:	21st Feb 2020
	Accepted:	25th Feb 2020

Editor: Lise Roth

Transaction Report:

1st Editorial Decision

14th Jun 2018

Thank you for the submission of your manuscript to EMBO Molecular Medicine. We have now heard back from the three referees whom we asked to evaluate your manuscript.

As you will see from the reports below, while the referees all mention the interest and potential clinical relevance of the study, they also agree that strengthening of the data to fully support the conclusions will be necessary for publication in EMBO Molecular Medicine. Our cross-commenting exercise helped in clarifying the most critical points to address during revision and we would encourage you to focus on the following:

- Provide mechanistic explanation of the notch-glucose relationship in vitro (referees #1 and #2)
- Analyse the kinetics of endothelial notch signaling after glucose ingestion or insulin stimulation in vivo (in wild-type and transgenic mice) to address the physiological significance and the direct role of insulin (referees #1 and #2)
- Perform Evans Blue experiments in vivo to rule out permeability issues (referees #2 and #3)
- Better present/explain the data and put them into context, especially with your recently published work (#1 and #3)

Addressing the above reviewers' concerns in full, and experimentally as needed, will be necessary for further considering the manuscript in our journal. EMBO Molecular Medicine encourages a single round of revision only and therefore, acceptance or rejection of the manuscript will depend on the completeness of your responses included in the next, final version of the manuscript.

***** Reviewer's comments *****

Referee #1 (Remarks for Author):

Summary:

Jabs and colleagues investigate the role of endothelial Notch signaling on systemic glucose metabolism. They describe that endothelial-restricted activation of Notch signaling decreases Insulin sensitivity thereby leading to increased blood glucose levels. Inhibition of Notch (by deletion of the common effector RBPj) causes opposite effects. As a mechanism of action, the authors propose that Notch signaling controls caveolae-dependent transport of insulin to skeletal muscle. This mechanism involves the transcriptional regulation of several caveolae genes, which are downregulated by Notch signaling.

Comment:

The manuscript describes several interesting metabolic phenotypes in endothelial-specific Notch gain- and loss-of-function mice, which suggest that endothelial Notch signalling plays a vital role in glucose homeostasis. While these observations appear robust, the molecular mechanism underlying these phenotypes is not convincing. As demonstrated, it remains unclear whether the Notch-modulated trans-endothelial transport of insulin is indeed responsible for the reported metabolic changes. Overall, the manuscript is still preliminary and fails to provide definitive evidence for the proposed mechanism.

Other comments:

Figure 1 - The mechanism by which high glucose levels induce increased NICD1 levels and Notch activity is not yet entirely apparent. The authors propose that a ROS-dependent activation of STAT3 leads enhanced Notch ligand expression, but the data supporting this model is weak or lacking. More details on the kinetics of Notch regulation as well as on the proposed molecular pathway would be necessary to understand the glucose-dependent modulation of Notch signaling.

Figure 2 - The metabolic data in Figure 2 are exciting and physiologically relevant. However, one wonders how these effects of altered Notch signaling on glucose metabolism relate to the effects on lipid metabolism recently described by the authors (Jabs et al., *Circulation* 2018).

The authors did not rule out that changes in endothelial Notch signaling alter the production of insulin or other hormonal regulators of glucose metabolism. Inhibition of Notch might lead to vascular alterations in the pancreas or liver, which might impact glucose levels.

The effects of Notch activation or inhibition of Insulin transport are an interesting observation. However, the effects appear relatively small, which makes one wonder whether this mechanism can account for the changes in blood glucose levels observed *in vivo*. The same accounts for the regulation of caveolae proteins, which are only mildly affected.

Does knockdown of the caveolar proteins regulated by Notch signaling suffice to impair insulin transport across the endothelium (*in vitro*)?

Figure 5a,b,e - The expression of the dnMAML1 and Notch1 ICD proteins needs to be shown.

Referee #3 (Remarks for Author):

This is an interesting study that elucidates the effect of Notch on endothelial transport of insulin. By using genetic mouse models, the authors demonstrate that activation of Notch reduces the sensitivity to insulin, whereas inhibition of Notch had the opposing effect. The authors also provide mechanistic insights into the pathways that are regulated by Notch to control insulin transport. The study is well performed and reports an interesting finding.

Major concern

The study overall is well performed, but more efforts should be invested to exclude unspecific effects of Notch on permeability. This is important, since Notch is a powerful EC regulator and changing the barrier function by toxic effects would confound the experiments. Thus, more control experiments such as staining for VE-cadherin to showing the integrity of the monolayer and assess TEER in the *in vitro* studies are essential.

Referee #4 (Remarks for Author):

The study by Jabs et al. investigates the relationship of Notch signaling, insulin and metabolic homeostasis. In essence they show that Notch signaling inhibits the transendothelial transport of insulin by a repressive transcriptional effect on caveolin proteins. They show in loss- and gain-of-function mouse models that Notch signaling affects insulin action *in vivo*, both at baseline in lean mice and under obesity-conditions. This exciting finding indicates that manipulation of the Notch pathway may have therapeutic potential for insulin resistance/diabetes. The results are novel and the experiments are well performed but there still weaknesses in this manuscript as outlined below.

Major concerns

1. Insulin action takes place within minutes after release from beta cells. In this study it remains unclear how the results obtained under chronic manipulation of Notch signaling are related to acute physiological changes in insulin levels in fasting and feeding cycles. The results in genetic obesity are interesting but remain anecdotal. How is Notch signaling in endothelial cells regulated after feeding? How does that impact Caveolae formation (timing from transcription to negative effect on insulin signaling)? Is this different for tissues like heart/skeletal muscle, white/brown adipose tissue etc. especially because these tissues vary greatly in their contribution to systemic metabolism? The paper is missing a strong foundation here.
2. The mechanistic Figure 1 is potentially relevant but in the end the panels are only loosely connected. The db/db mouse model has many more metabolic derangements other than just hyperglycemia and hyperinsulinemia so the findings are in line but do not really corroborate the conclusion. In the cells, how is glucose linked to Notch signaling? Where are the natural Notch ligands coming from? The conclusion that ROS-Stat3 signaling is involved is premature and should be more carefully explored for a more solid foundation of the study, particularly the potential link between glucose uptake, Stat3 activation and Notch signaling.
3. More generally, the interpretation that Notch activation by glucose is a feedback inhibition of insulin transport is interesting but some clarification regarding whether this has actually physiologic significance under more dynamic circumstances would be supportive.

Minor points

4. In general, in some panels, immunoblot quantifications are shown without the images (e.g. Fig. 3f,g) and in some the images are shown without quantification (e.g. Fig. 1b, Fig. 5a etc.). I recommend that all blots are quantified, uncropped images are included as supplementary material and that the authors indicate how many times the blots were performed.
5. Some expert readers will be judgmental about the absence of hyperinsulinemic-euglycemic clamp studies. While I certainly think they would be helpful here, also for assessing tissue-specific effects and glucose uptake, I recommend that the authors always show both glucose tolerance test (GTT) as well as insulin tolerance tests (ITT), as currently in Fig. 2 there are only ITTs and in Fig. 6 there are only GTTs shown. My personal opinion is that ITTs should not be shown as percentage of 0 min but rather as absolute plasma glucose concentrations. Also, fasting as well as fed (e.g. during GTT) plasma insulin levels are required for all *in vivo* studies.
6. I recommend that also the raw data for experiments, for which quantifications are shown, are included in the paper (e.g. Fig 4c the actual images).
7. For the inducible KO models, some quality control measures like endothelial mRNA or protein expression of the deleted protein of interest should be included as well.
8. In Fig. 3b for example the error bar is missing. For fold calculations, the error is STD/SEM divided by mean/average multiplied by fold.
9. The authors say there is no effect on barrier function but at least actually testing this, also in light of the recent JCI paper by Williams et al. 2018 is recommended (e.g. by Evans Blue injections)

10. In experiment Fig. 3d the result with Filipin is difficult to interpret because the control condition of no dnMAML1 with Filipin treatment is missing.

11. For the immunofluorescence pictures I recommend showing the individual channels in black and white for better visibility and interpretation.

12. If the authors have Notch-manipulating tools like antibodies in hand, it would be great to add some "medicine" to the paper like treating diet-induced mice with Notch-manipulating agents and look at insulin action/insulin resistance.

13. Given that the authors have observed significant metabolic changes in the mouse models used in this study (Jabs et al Circulation 2018), the obvious question is how do these studies relate to each other?

1st Revision - authors' response

31st May 2019

Please see next page.

Referee #1:

The manuscript describes several interesting metabolic phenotypes in endothelial-specific Notch gain- and loss-of-function mice, which suggest that endothelial Notch signaling plays a vital role in glucose homeostasis. While these observations appear robust, the molecular mechanism underlying these phenotypes is not convincing. As demonstrated, it remains unclear whether the Notch-modulated trans-endothelial transport of insulin is indeed responsible for the reported metabolic changes. Overall, the manuscript is still preliminary and fails to provide definitive evidence for the proposed mechanism.

We thank the reviewer for the positive evaluation of our work. The comments helped us to improve the manuscript extensively. We have substantially revised the manuscript and we have re-written several parts to streamline it and to explain the proposed mechanism better.

To address the concern regarding the mechanism: we alter Notch signaling activity in all endothelial cells and therefore it is of course not possible to nail down a systemic phenotype to one single mechanism in one single organ. Here we have focused on muscle. The data clearly indicate that loss of endothelial Notch leads to an enhanced insulin sensitivity. This is based on measuring glucose and insulin at basal rates (both are decreased) as well as classical insulin and glucose tolerance tests. As also outlined below, this is very unlikely to be an insulin production problem, as in this case, one would expect low insulin but high plasma glucose. Furthermore, the experiment that measures pAKT in muscles upon i.v. insulin injection shows higher phosphorylation of AKT, indicating a faster and/or increased insulin transport across the vessel wall to muscle cells. This might be due to increased transendothelial transport or increased permeability. We have addressed both the possibilities below.

Transendothelial insulin transport via caveolae is well described in particular in muscle tissue. In other organs, there are different ways to shuttle insulin across the vessel wall. For example in liver, the sinusoidal endothelium provides gaps for free diffusion of insulin towards hepatocytes. In our study, both the cell culture models and the data obtained from mouse tissues show that upon loss of endothelial Notch signaling there are high expression rates of genes critically involved in caveolae formation, high numbers of caveolae in endothelial cells, as well as better insulin uptake into endothelial cells.

Regarding permeability and perfusion, we could show in our previous work (Wieland et al., Cancer Cell 2017 and Jabs et al., Circulation 2018) that manipulation of endothelial Notch signaling increases permeability in endothelial cell culture transwell models. However, *in vivo* we did not observe any signs of vascular leakage. In addition, tracer studies, even in severe inflammation models, did not show an increase in vessel permeability. Actually, we observed a slight decrease in tracer flux (please refer to some of the published data sets below). As such, our proposed model of increased insulin flux across the vessel wall to muscle cells due to altered caveolae formation is the most likely mechanism for the observed effects.

In addition, we had previously observed that the heart consumes more glucose upon endothelial Notch deletion as a compensatory mechanism because of impaired fatty acid transport. This may contribute to the lowered basal glucose levels. Nevertheless, the GTT and ITT data in conjunction with the increased pAKT in skeletal muscle argue for enhanced insulin transport also in skeletal muscle.

Lastly, we would like to mention that published mouse models of caveolin-1 knockout interfering with caveolae formation and expression of caveolin genes nicely fit to our study.

Other comments:

Figure 1 - The mechanism by which high glucose levels induce increased NICD1 levels and Notch activity is not yet entirely apparent. The authors propose that a ROS-dependent activation of STAT3 leads enhanced Notch ligand expression, but the data supporting this model is weak or lacking. More details on the kinetics of Notch regulation as well as on the proposed molecular pathway would be necessary to understand the glucose-dependent modulation of Notch signaling.

We performed extensive additional experiments to address this issue further. Several groups have reported activation of Notch signaling (based on transcriptional profile of Notch components and target genes) under high glucose conditions *in vitro*. We found that this effect could only be (partially) confirmed when glucose (5g/L) was added together with fresh media (see panels below). However, adding glucose directly to the medium of primary human macro and microvascular endothelial cells (HUVECs, HDMECs) as well as mouse cardiac endothelial cells (MCECs) did not lead to robust Notch induction in particular when the cleavage rate of Notch1 was determined. We performed these experiments using various control conditions including low glucose (1g/L) combined with Mannitol for comparable osmolarity and L-Glucose (non-metabolizable, 5g/L).

It is most likely that the very strong induction rate initially observed is due to factors in cell culture medium such as BMP9 that strongly activate *Hes* and *Hey* gene transcription in a

Notch independent manner. We have reported this effect in our previous study (Wöltje et al., PLoS One 2015 Mar 23;10(3):e0120547).

Nevertheless, in endothelial cells derived from obese db/db mice there was increased Notch signaling activity (Fig. 1A). This is similar to increased Notch activity observed in retinal endothelial cells in a Streptozotocin mouse model of type I diabetes mellitus (Miloudi et al., PNAS 2019). However, fasting-feeding cycles were not sufficient to induce endothelial Notch signaling activity significantly (Fig. 1B). Therefore, it appears very likely that only a chronic disturbance in metabolism as seen in obesity or severe type I diabetes robustly induces Notch signaling activity in endothelial cells. Therefore, our genetic model resembles the effect of chronically disturbed metabolism and not the physiological metabolic changes that occur frequently every day.

Figure 2 - The metabolic data in Figure 2 are exciting and physiologically relevant. However, one wonders how these effects of altered Notch signaling on glucose metabolism relate to the effects on lipid metabolism recently described by the authors (Jabs et al., Circulation 2018).

The metabolic data are now shown in the new Figures 1 and 2. We have now added a paragraph in the Discussion to better relate our results to the changes in glucose and lipid metabolism described by us recently (Jabs et al., Circulation 2018). As outlined above, the increased glucose uptake into cardiomyocytes might contribute to the lowered blood glucose in endothelial cell-specific *Rbpj*-deficient mice. However, the enhanced insulin transport across the endothelium with subsequent increased glucose uptake and consumption is most likely an important mechanism why the heart takes up more glucose and that this is not only due to compensation because of impaired fatty acid transport.

Combining data from this manuscript and our previously published work (Jabs et al., Circulation 2018), we can conclude that Notch signaling plays a role in balancing fatty acid uptake and glucose metabolism under physiological conditions.

The authors did not rule out that changes in endothelial Notch signaling alter the production of insulin or other hormonal regulators of glucose metabolism. Inhibition of Notch might lead to vascular alterations in the pancreas or liver, which might impact glucose levels.

Following the reviewer's advice, we performed stainings on pancreas section for both Notch loss of function and gain of function mice. We did not detect any significant differences in vessel area and pancreatic islet area in *NICD^{IOE-EC}* mice (Fig EV1F-I). We also did not detect any changes in pancreatic islet area in *Rbpj^{ΔEC}* mice. We did however observe an increase in vessel area in *Rbpj^{ΔEC}* mice (Fig EV2B-E). This is similar to what we had observed in muscle tissue (Jabs et al., Circulation 2018) and indicates once more that endothelial Notch signaling is needed throughout adulthood to inhibit new blood vessel growth. However, as described in our previous study we did not see any increase in vessel permeability (see panels below from Supplementary Fig. 3, Jabs et al., Circulation 2018).

The effects of Notch activation or inhibition of Insulin transport are an interesting observation. However, the effects appear relatively small, which makes one wonder whether this mechanism can account for the changes in blood glucose levels observed in vivo. The same accounts for the regulation of caveolae proteins, which are only mildly affected. Does knockdown of the caveolar proteins regulated by Notch signaling suffice to impair insulin transport across the endothelium (in vitro)?

We did perform *Cav1* knock-down in transwell assays (Fig 4A-D) to investigate insulin transport across endothelial monolayer to muscle cells. Indeed, *Cav1* knockdown in ECs was sufficient to impair insulin transport. As our model reflects chronic changes in metabolism followed by chronic changes in Notch signaling activity, such rather small changes in caveolar proteins and caveolae number would still accumulate to strong changes over longer time periods.

The slight decrease of *Cav1* might indeed impair insulin transport in a much stronger manner compared to a complete loss of *Cav1* as this scenario opens paracellular junctions and increases permeability (Wang et al., AJEPM 2011 and Schubert et al., JBC 2002).

Figure 5a,b,e - The expression of the dnMAML1 and Notch1 ICD proteins needs to be shown.

These constructs are well-established tools in our laboratory, which have been published in previous papers by many laboratories. Since the constructs are fragments of the endogenous proteins (the short dominant-negative version of MAML1), we cannot show the expression on the protein level, as the antibodies cannot detect the remaining protein fragment. However, we now provide data showing that the constructs regulate Notch target gene expression on the mRNA level as expected (Fig EV3B).

Referee #3:

This is an interesting study that elucidates the effect of Notch on endothelial transport of insulin. By using genetic mouse models, the authors demonstrate that activation of Notch reduces the sensitivity to insulin, whereas inhibition of Notch had the opposing effect. The authors also provide mechanistic insights into the pathways that are regulated by Notch to control insulin transport. The study is well performed and reports an interesting finding.

We thank the reviewer for the positive evaluation of our work. The comments helped us to improve the manuscript substantially.

Major concern:

The study overall is well performed, but more efforts should be invested to exclude unspecific effects of Notch on permeability. This is important, since Notch is a powerful EC regulator and changing the barrier function by toxic effects would confound the experiments. Thus, more control experiments such as staining for VE-cadherin to showing the integrity of the monolayer and assess TEER in the *in vitro* studies are essential.

We completely agree with the reviewer's concern. We have shown in our previous works (Wieland et al., Cancer Cell 2017 and Jabs et al., Circulation 2018) that manipulation of endothelial Notch signaling increases permeability in endothelial cell culture transwell models. As the reviewer suggested, we had already determined VE-Cadherin (impaired expression at cell-cell contacts) and TEER (see below). Keeping these results in mind, we have not included any *in vitro* data in the revised manuscript regarding transendothelial insulin flux. Now, we only show the cellular uptake of insulin (Fig 3A, B).

Importantly, we did not observe any signs of vascular leakage *in vivo*. In addition, tracer studies, even in severe inflammation models, did not show an increase in vessel permeability. Actually, we observed a slight decrease in tracer flux (please refer to some of the published data sets below). As such, our proposed model of increased insulin flux across the vessel wall to muscle cells due to altered caveolae formation is the most likely mechanism for the observed effects.

Figure 5G and H from Wieland et al., Cancer Cell 2017

Endothelial Notch Activation Promotes TC Adhesion and Transmigration

Loss of Notch signaling increases permeability of endothelial *in vitro* but not *in vivo*.

 Referee #4:

The study by Jabs et al. investigates the relationship of Notch signaling, insulin and metabolic homeostasis. In essence they show that Notch signaling inhibits the transendothelial transport of insulin by a repressive transcriptional effect on caveolin proteins. They show in loss- and gain-of-function mouse models that Notch signaling affects insulin action *in vivo*, both at baseline in lean mice and under obesity-

conditions. This exciting finding indicates that manipulation of the Notch pathway may have therapeutic potential for insulin resistance/diabetes. The results are novel and the experiments are well performed but there still weaknesses in this manuscript as outlined below.

We thank the reviewer for the positive evaluation of our work. The comments helped us to improve the manuscript substantially.

Major concerns

1. Insulin action takes place within minutes after release from beta cells. In this study it remains unclear how the results obtained under chronic manipulation of Notch signaling are related to acute physiological changes in insulin levels in fasting and feeding cycles. The results in genetic obesity are interesting but remain anecdotal. How is Notch signaling in endothelial cells regulated after feeding?

To distinguish between a chronic and acute response to alterations in plasma metabolites, we analyzed Notch targets in primary ECs freshly isolated from hearts of fasted and refed mice. We observed no significant differences in Notch target gene expression in ECs from mice fasted overnight when compared to ECs from mice, which were refed after fasting (Fig 1B). However, chronic changes in plasma metabolites such as during the course of obesity in db/db mice increased Notch signaling in endothelial cells (Fig. 1A). Therefore, our mouse models with chronically increased or decreased endothelial Notch signaling fit nicely to this situation.

How does that impact Caveolae formation (timing from transcription to negative effect on insulin signaling).

Since the feeding/fasting experiment did not show an effect on Notch signaling we also do not expect an effect on caveolae formation in this rather short time point. As outlined above, this study addresses a long-term change of endothelial Notch signaling activity as e.g. seen in obese mice. After gene recombination, it takes approximately four weeks to observe significant changes in blood glucose levels.

Is this different for tissues like heart/skeletal muscle, white/brown adipose tissue etc. especially because these tissues vary greatly in their contribution to systemic metabolism? The paper is missing a strong foundation here.

This is a very interesting question. This manuscript addresses only heart and skeletal muscle. The literature indicates that insulin transport differs substantially between organs. Analysis of other major regulatory organs (brain, liver, WAT, BAT) goes far beyond the scope of this manuscript. We are currently addressing some of these questions and it has become apparent that the consequence of endothelial Notch signaling is strictly organ-specific.

2. The mechanistic Figure 1 is potentially relevant but in the end the panels are only loosely connected. The db/db mouse model has many more metabolic derangements other than just hyperglycemia and hyperinsulinemia so the findings are in line but do not really corroborate the conclusion. In the cells, how is glucose linked to Notch signaling? Where are the natural Notch ligands coming from? The conclusion that ROS-Stat3 signaling is involved is premature and should be more carefully explored for a more solid foundation of the study, particularly the potential link between glucose uptake, Stat3 activation and Notch signaling.

We agree that db/db mice have a large repertoire of metabolic alterations. We have discussed this now in the revised manuscript. Regarding the link between high glucose and Notch we performed extensive additional experiments to address this issue. Several groups have reported activation of Notch signaling (based on transcriptional profile of Notch components and target genes) under high glucose conditions *in vitro*. We found that this effect could only be (partially) confirmed when glucose (5g/L) was added together with fresh media (see panels below). However, adding glucose directly to the medium of primary human macro and microvascular endothelial cells (HUVECs, HDMECs) as well as mouse cardiac endothelial cells (MCECs) did not lead to robust Notch induction in particular when the cleavage rate of Notch1 was determined. We performed these experiments using various control conditions including low glucose (1g/L) combined with Mannitol for comparable osmolarity and L-Glucose (5g/L). Therefore, more sophisticated downstream analysis of e.g. Stat3 did not make sense anymore.

HDMECs High Glucose Treatment 24h

HDMECs High Glucose Treatment 48h

HUVECs High Glucose Treatment

HUVECs High Glucose Treatment

It is most likely that the very strong induction rate initially observed is due to factors in cell culture medium such as BMP9 that strongly activates *Hes* and *Hey* gene transcription in a Notch independent manner. We have reported this effect in our previous study (Wöltje et al., PLoS One 2015 Mar 23;10(3):e0120547.).

Nevertheless, in endothelial cells derived from obese db/db mice there was increased Notch signaling activity (Fig. 1A). This is similar to increased Notch activity observed in retinal endothelial cells in Streptozotocin mouse model of type I diabetes mellitus (Miloudi et al., PNAS 2019). However, fasting-feeding cycles were not sufficient to induce endothelial Notch

signaling activity significantly (Fig. 1B). Therefore, it appears very likely that only a chronic disturbance in metabolism as seen in obesity or type I diabetes robustly induces Notch signaling activity in endothelial cells. Therefore, our genetic model resembles the effect of chronically disturbed metabolism and not the physiological metabolic changes that occur frequently every day.

Regarding the natural source of Notch ligand, we saw an increase of both Notch ligands Jag1 and Dll4 in endothelial cells from db/db mice. In the resting endothelium, the general notion is that Notch signaling occurs at cell-cell contacts within the endothelium.

3. More generally, the interpretation that Notch activation by glucose is a feedback inhibition of insulin transport is interesting but some clarification regarding whether this has actually physiologic significance under more dynamic circumstances would be supportive.

As outlined in the answer to question 1 we see the induction of Notch due to a chronic stimulation during the course of obesity. This would subsequently contribute to insulin resistance. As such, the manuscript is not addressing a physiological response mechanism but rather a mechanism that contributes to disease progression.

Minor points

4. In general, in some panels, immunoblot quantifications are shown without the images (e.g. Fig. 3f, g) and in some the images are shown without quantification (e.g. Fig. 1b, Fig. 5a etc.). I recommend that all blots are quantified, uncropped images are included as supplementary material and that the authors indicate how many times the blots were performed.

We have quantified all the western blots in the manuscript and included the quantifications in the figures. Furthermore, we have provided original blots in the source data set.

5. Some expert readers will be judgmental about the absence of hyperinsulinemic-euglycemic clamp studies. While I certainly think they would be helpful here, also for assessing tissue-specific effects and glucose uptake, I recommend that the authors always show both glucose tolerance test (GTT) as well as insulin tolerance tests (ITT), as currently in Fig. 2 there are only ITTs and in Fig. 6 there are only GTTs shown.

We have now included GTTs to the ITTs for both NICD^{iOE-EC} mice and *Rbpj*^{ΔEC} mice (Fig 1 and 2). Unfortunately, we could not repeat the ITTs for the high fat diet experiment as it would require repeating the experiments for which we would require additional clearance from the Animal Ethics Committee.

My personal opinion is that ITTs should not be shown as percentage of 0 min but rather as absolute plasma glucose concentrations.

We have amended this.

Also, fasting as well as fed (e.g. during GTT) plasma insulin levels are required for all in vivo studies.

Since, this is not a standard practice we did not measure plasma insulin levels. In addition, our protocol approved by the Animal Ethics Committee does not allow taking large quantities of blood to measure both.

6. I recommend that also the raw data for experiments, for which quantifications are shown, are included in the paper (e.g. Fig 4c the actual images).

We have included the images for the proximity ligation assay. In addition, we have changed

the data presentation whenever possible, so we now show the real data points throughout all figures.

7. For the inducible KO models, some quality control measures like endothelial mRNA or protein expression of the deleted protein of interest should be included as well.

These animal models are very well established in the field and had been validated in our previous study (see panel below). Furthermore, as a standard lab practice we always perform PCRs to check for gene recombination on all animals used in experiments.

Supplementary Fig 1 from Jabs et al., Circulation 2018

Endothelial-specific ablation of Rbpj- κ in adult mice.

8. In Fig. 3b for example the error bar is missing. For fold calculations, the error is STD/SEM divided by mean/average multiplied by fold.

The manuscript does not contain the Fig. 3b anymore as the revised manuscript does not contain any transwell assays with Notch manipulation due to permeability issues. For all other figures, data points and error bars are shown.

9. The authors say there is no effect on barrier function but at least actually testing this, also in light of the recent JCI paper by Williams et al. 2018 is recommended (e.g. by Evans Blue injections)

We completely agree with the reviewer's concern. We have shown in our previous works (Wieland et al., Cancer Cell 2017 and Jabs et al., Circulation 2018) that manipulation of endothelial Notch signaling increases permeability in endothelial cell culture transwell models (see below). Keeping these results in mind, we have not included any *in vitro* data in the revised manuscript regarding transendothelial insulin flux. Now, we only show the cellular uptake of insulin (Fig 2A, B).

However, we did not observe any signs of vascular leakage *in vivo*. In addition, tracer studies, even in severe inflammation models, did not show an increase in vessel permeability, actually, we observed a slight decrease in tracer flux (please refer to some of the published data sets below). As such, our proposed model of increased insulin flux across the vessel wall to muscle cells due to altered caveolae formation is the most likely mechanism for the observed effects.

Figure 5G and H from Wieland et al., Cancer Cell 2017

Endothelial Notch Activation Promotes TC Adhesion and Transmigration

Supplementary Fig 3 from Jabs et al., Circulation 2018

Loss of Notch signaling increases permeability of endothelial *in vitro* but not *in vivo*.

10. In experiment Fig. 3d the result with Filipin is difficult to interpret because the control condition of no dnMAML1 with Filipin treatment is missing.

This experiment is not included in the manuscript anymore as the changes in permeability upon Notch manipulation *in vitro* might affect paracellular insulin flux. Instead, we only show insulin uptake in endothelial cells upon Notch manipulation (Fig. 3A, B)

11. For the immunofluorescence pictures I recommend showing the individual channels in black and white for better visibility and interpretation.

We have improved all immunofluorescence pictures to better show anatomical structures and co-localization of proteins using the classical red/green/yellow approach.

12. If the authors have Notch-manipulating tools like antibodies in hand, it would be great to add some "medicine" to the paper like treating diet-induced mice with Notch-manipulating agents and look at insulin action/insulin resistance.

This is an excellent suggestion. Unfortunately, we could not receive an MTA with a pharma company to do such experiments.

13. Given that the authors have observed significant metabolic changes in the mouse models used in this study (Jabs et al Circulation 2018), the obvious question is how do these studies relate to each other?

The metabolic data are now shown in the new Figures 1 and 2. We have now added a paragraph in the Discussion to better relate our results to the changes in glucose and lipid metabolism described by us recently (Jabs et al., Circulation 2018). As outlined above, the increased glucose uptake into cardiomyocytes might contribute to the lowered blood glucose in endothelial cell-specific Rbpj-deficient mice. However, the enhanced insulin transport across the endothelium with subsequent increased glucose uptake and consumption is most likely an important mechanism why the heart takes up more glucose and that this is not only due to compensation because of impaired fatty acid transport.

Combining data from this manuscript and our previously published work (Jabs et al., Circulation 2018), we can conclude that Notch signaling plays a role in balancing fatty acid uptake and glucose metabolism under physiological conditions.

Thank you for the resubmission of your manuscript to EMBO Molecular Medicine, and please accept my apologies for the delay in getting back to you, which is due to the fact that one referee needed more time to complete his/her review. We have now received feedback from the three experts who had initially reviewed the first version of your manuscript. As you will see from the reports below, while referee #3 is satisfied with the revisions and supports publication of the manuscript, referee #1 and #4 still have serious concerns, particularly regarding the reported mechanism that needs to be further strengthened.

As EMBO Press encourages a single round of revisions only, and given the referees' remaining concerns, we would normally reject the manuscript at this stage. However, we decided to exceptionally allow a second round of revisions. Please be aware that this will be the last chance for you to address all the points raised by the referees, and that acceptance or rejection of the manuscript will depend on the completeness of your responses included in the next, final version of the manuscript. For this reason, and to save you from any frustrations in the end, I would strongly advise against returning an incomplete revision.

***** Reviewer's comments *****

Referee #1 (Remarks for Author):

Jabs and colleagues have resubmitted a revised version of their manuscript "Endothelial Notch signaling controls insulin transport in muscle" that contains further clarifications, new experiments, and data. These new additions address some of my previous concerns, while others remain. Although the manuscript describes interesting observations, it still lacks rigor and experimental consistency

1. It is still not clear whether endothelial Notch signaling directly regulates insulin transport through the endothelium. The authors use several approaches to try to make this point. However, all of them are indirect in nature and do not provide unambiguous evidence for the proposed mechanism. For instance, Akt phosphorylation, which the authors use as a readout for insulin transfer through the endothelium, is not specific for insulin and could be the result of many any signaling events. Measuring AKT phosphorylation in muscle lysate is even less, and it remains unclear which conclusions can be made from these experiments. As a minimum, the authors should try to measure insulin transport in cultured endothelial cells. Co-culture studies with labeled insulin should be able to make this point.

2. The authors are inconsistent with the use of their models, and they jump back and forth between analyzed tissues (e.g. heart, lung, skeletal muscle) and ways to alter Notch signaling (e.g. Dll4, dnMAML, NICD). To make a convincing case for the proposed mechanism, they should make a thorough effort to focus their analyses on skeletal muscle using consistent tools.

3. The authors provide some data that suggest that overall pancreas function is not affected; however, they did not exclude other relevant perturbations. For instance, Notch inhibition is known to cause liver pathology (e.g. Yan et al. Nature 2006), which might affect general glucose homeostasis. Excluding major changes in liver function, thus, appears mandatory.

4. The mechanism by which endothelial Notch signaling regulates the expression of the caveolar proteins is still unclear. To make a more convincing case, the authors should assess whether CAV1, CAV2, and CAVIN1 are direct Notch target gene.

5. Figure 3C: The authors should provide the unphosphorylated AKT immunoblot as a control. Same accounts for Figure 3E.

6. Figure 5B: The levels of NICD overexpression need to be shown.

Referee #3 (Remarks for Author):

The authors adequately addressed my concerns.

Referee #4 (Comments on Novelty/Model System for Author):

The performed experiments are well done. The results are novel. The authors show that manipulation of their pathway impacts metabolic health in obesity mouse models.

Referee #4 (Remarks for Author):

The revised version by Jabs et al. has significantly improved. The data are now presented in the appropriate way and the quality of the study is greatly enhanced. However, two critical major questions still remain completely unaddressed in the revised version:

Major concerns:

1. Still, there is not a single figure in the paper showing a difference in muscle glucose uptake (neither was insulin uptake itself measured), which would ultimately proof the concept that it is actually insulin-stimulated glucose uptake into muscle that is underlying or at least contributing to the phenotype of the transgenic animals with manipulation of Notch signaling. The *in vitro* data on caveolin and insulin receptor are very nice, but we don't know whether there are differences in insulin receptor availability in muscle/myocytes actually translating into glucose uptake *in vivo*. The authors show insulin signaling in muscle and liver but not in other tissues, where insulin needs to cross the barrier. Still, it is possible that the phenotype *in vivo* is independent of muscle insulin action and muscle glucose uptake and there is a lot of faith required to acknowledge the concept as suggested. Ideally, as I mentioned before, this should be confirmed by clamp studies, in which radiolabeled tracers are used to follow the fate of glucose into the respective tissues. Alternatively, a glucose tolerance test (the authors only assume that postprandial insulin is higher but don't show that - 0 and 10 min blood would be required for that), in which ¹⁴C-deoxyglucose or other tracers are mixed in, would allow to proof their concept. Higher postprandial insulin with lower glucose uptake would prove their concept. Even if they find that this holds also true in other tissues like white or brown fat etc. it would make the study stronger with a broader scope. I do not want to stress this, but I want to mention that the islet phenotype calls for primary islet insulin release assays, as the differences in vascularization might impact insulin secretion somehow. Showing the impact of Notch signaling on *in vivo* glucose (insulin?) muscle uptake is imperative for this kind of metabolic study, especially with the strong claims made here.
2. The paper claims that Notch signaling controls insulin action in muscle, yet there is no physiological regulation of the Notch signaling with fasting and feeding and, therefore, the study does not really have a physiological scope and now has a very different angle. One could argue that Notch signaling constitutively inhibits insulin action through a repression of caveolae formation (why are there no muscle ECs data in Figure 5?). This does not seem to be a developmental defect, and this is an important finding by itself. The authors argue that in the db/db model, Notch signaling is enhanced (Why do the authors show lung ECs, and not muscle ECs - in the context of my point #1 brings me back to the question of how much muscle is in the *in vivo* phenotype or is this systemic - heart we apparently know already...), which might contribute to the insulin resistance in this model somehow. They also show that in the HFD model, inducible deletion of Notch signalling improves glucose tolerance independently of body weight (Fig. 6), which is a very strong experiment and the findings are really exciting. Altogether, this gives this study a more pathophysiological obesity-linked angle. However, there are two major complications (and the *in vitro* high glucose incubations are not convincing): The db/db model a priori is a leptin signaling-deficient model so whether the changes in EC Notch signaling are due to leptin deficiency or due to the metabolic disease in this model cannot not be deduced and this finding remains anecdotal. Much better would be showing muscle ECs from lean and HFD mice next to db/db and controls with some stronger readout than qPCR - I think given that this study now has an obesity focus and the authors make very strong claims based on very little data this is unacceptable. The other is that in the HFD inducible deletion experiment, insulin levels are not shown, ITT was not performed, and muscle glucose uptake was

not measured by any means. Altogether, this emphasizes that the authors are not experts in metabolic experiments, but I think their basic concept, particularly as it seems to have some therapeutic benefit for obesity-induced insulin resistance, is still very exciting.

2nd Revision - authors' response

3rd Feb 2020

Please see next page.

Referee #1 (Remarks for Author):

Jabs and colleagues have resubmitted a revised version of their manuscript "Endothelial Notch signaling controls insulin transport in muscle" that contains further clarifications, new experiments, and data. These new additions address some of my previous concerns, while others remain. Although the manuscript describes interesting observations, it still lacks rigor and experimental consistency.

We thank the reviewer for the positive evaluation of our work. The comments helped us to improve the manuscript extensively. We have substantially revised the manuscript according to the reviewers' comments.

1. It is still not clear whether endothelial Notch signaling directly regulates insulin transport through the endothelium. The authors use several approaches to try to make this point. However, all of them are indirect in nature and do not provide unambiguous evidence for the proposed mechanism. For instance, Akt phosphorylation, which the authors use as a readout for insulin transfer through the endothelium, is not specific for insulin and could be the result of many any signaling events. Measuring AKT phosphorylation in muscle lysate is even less, and it remains unclear which conclusions can be made from these experiments. As a minimum, the authors should try to measure insulin transport in cultured endothelial cells. Co-culture studies with labeled insulin should be able to make this point.

We agree with the reviewer's insightful comments. In our previous version of the manuscript, we did include a transwell assay for insulin transport across cultured endothelial cells to muscle cells that shows decreased endothelial insulin flux upon Notch stimulation and increased insulin flux upon Notch inhibition (see below).

However, we removed them from the later version due to concerns regarding the permeability of endothelial cells after Notch manipulation as suggested by one of the other referees. We have shown in our previous publications (Wieland et al., Cancer Cell 2017 and Jabs et al., Circulation 2018) that manipulation of endothelial Notch signaling increases permeability in endothelial cell culture transwell models, but not in mice (see below). Keeping these results in mind, we have not

included any *in vitro* data in the revised manuscript regarding transendothelial insulin flux, as the disturbance of endothelial permeability precludes any judgement on paracellular vs transcellular flux. Now, we only show the cellular uptake of insulin in endothelial cells upon Notch manipulation, which is independent of any permeability issues (Fig 3A, B).

Figure 5G and H from Wieland et al., Cancer Cell 2017
Endothelial Notch Activation Promotes TC Adhesion and Transmigration

Supplementary Fig 3 from Jabs et al., Circulation 2018
Loss of Notch signaling increases permeability of endothelial *in vitro* but not *in vivo*.

We have also commented on this issue in the discussion section in the manuscript (see below)

“In addition, a recent study has demonstrated that non-canonical Notch signaling limits vascular permeability³⁴. In previous studies from our laboratory, we had observed increased permeability in NICD-expressing HUVEC monolayer²⁷, and Notch inhibition increased paracellular permeability as determined by trans-endothelial electrical resistance and capacity measurements in HUVECs²⁰. Therefore, we did not include any in vitro experiments to measure insulin flux across endothelial monolayer upon Notch manipulation in this study. However, in vivo, as described previously²⁰, we did not see any increase in extravasation of Evans blue between Rbpj^{ΔEC} mice and corresponding littermate controls. Hence, the increased insulin flux across the endothelium observed in Rbpj^{ΔEC} mice is most likely not due to grossly altered paracellular permeability.”

Regarding pAKT as readout for insulin signaling *in vivo* in muscle tissues, we agree that other signaling pathways can affect AKT phosphorylation. However, since in our case we hardly detect any pAKT with saline injections, we could assume that the observed increase in pAKT levels just a few minutes later is a very valid readout of insulin action. Furthermore, this is a very well established assay, used in numerous publications to show insulin action, also in skeletal muscles. In addition, to strengthen our hypothesis, we have now also included glucose uptake assays in insulin responsive tissues like muscles (Fig 6, see below).

Lastly, we would like to mention that our very strict animal welfare regulations do not allow repeating such an *in vivo* experiment that was already successfully done before.

Fig 6

C

D

H

I

2. The authors are inconsistent with the use of their models, and they jump back and forth between analyzed tissues (e.g. heart, lung, skeletal muscle) and ways to alter Notch signaling (e.g. Dll4, dnMAML, NICD). To make a convincing case for the proposed mechanism, they should make a thorough effort to focus their analyses on skeletal muscle using consistent tools.

We fully understand the concerns of the reviewer. We have now performed new experiments exclusively with endothelial cells freshly isolated from skeletal muscle tissue to replace e.g. data using lung endothelial cells or cardiac muscle endothelial cells. The new data include analysis of skeletal muscle endothelial cells for Notch targets in obese mice (Fig 1A, see below) and during physiological fasting/feeding cycles (Fig 1B, see below). We also analyzed caveolar component genes in skeletal muscle endothelial cells in both Notch loss and gain of function mouse models and obese mice (Fig 5G-I, see below). Importantly, these data confirm our previous findings.

Fig 1

Fig 5

Lastly, we would like to mention that these analyses were accompanied by a few assays using HUVECs, which is the prototypical primary endothelial cell type, used in the vast majority of studies in the vascular biology field. The reason for this is that HUVECs can be easily genetically modified and can be cultured for a much longer period compared to isolated skeletal muscle cells which do not proliferate in culture any more.

For the cell culture experiments, we have used expression of active Notch1 (NICD) to induce Notch signaling or dnMAML1 to inhibit Notch signaling. Both of these approaches are very well established and we have included data in Fig EV5B (see below) to show their efficacy. Only one experiment (insulin uptake in cultured endothelial cells) was performed with coated DLL4 to induce Notch activity. Again, this is a classical way to stimulate Notch signaling. The reason we used this instead to induce Notch signaling is the fact that the NICD viral construct contains a GFP cassette that leads to very weak but still detectable fluorescence. As we used FITC-insulin fluorescence as readout we could not use the NICD construct, therefore we decided to use coated DLL4.

Nevertheless, we would like to mention that research in the Notch field is usually done by using multiple tools (including soluble ligands, Rbpj-VP16 or gamma secretase inhibitors) and that different experiments are shown using suitable tools based on experiment design.

Fig EV5B

3. The authors provide some data that suggest that overall pancreas function is not affected; however, they did not exclude other relevant perturbations. For instance, Notch inhibition is known to cause liver pathology (e.g. Yan et al. Nature 2006), which might affect general glucose homeostasis. Excluding major changes in liver function, thus, appears mandatory.

We have now tested multiple parameters for liver function and morphology in our endothelial specific *Rbpj^{ΔEC}* mouse model. We did see an increase in microvessel density and sinusoidal dilation in the liver sections in these mice as it was published before by others (Cuervo, H. *et al*, Hepatology, 2016). Most importantly, we did not detect any hepatic necrosis, fibrosis or iron deposition in histological sections (Fig EV4C-E, see below). In addition, albumin, urea, transaminases and alkaline phosphatase levels were within the normal limits in plasma from both the groups (Fig EV4F-J, see below). Therefore, we can rule out major changes in liver function that would affect systemic glucose homeostasis.

Fig EV5

4. The mechanism by which endothelial Notch signaling regulates the expression of the caveolar proteins is still unclear. To make a more convincing case, the authors should assess whether *CAV1*, *CAV2*, and *CAVIN1* are direct Notch target gene.

We would like to apologize to the reviewer if we have not explained our hypothesis in a better way. We are not proposing that *CAV1*, *CAV2* and *CAVIN1* are direct Notch targets. If that were the case, we would see an upregulation of these genes upon Notch induction. On the contrary, we see a downregulation of these genes upon Notch induction both *in vitro* and *in vivo* (Fig 5, see below), which points towards repression via the classical Notch target genes of the Hes and/or Hey transcriptional repressors.

Fig 5

Notch signaling often works through transcriptional activation of Hey/Hes family of transcriptional repressors that in turn transcriptionally downregulate target genes. In our case, we propose that Notch signaling downregulates expression of caveolar genes through the expression of Hey/Hes. Previous studies have shown that Hey1 physically interacts with the Cav1 promoter thereby repressing it. We have included data that shows that HEY1 overexpression in HUVECs downregulates *CAV1* expression (Fig 5F, see below). Furthermore, HEY1 overexpression also

significantly lowers the upregulation of *CAV1* that is observed upon Notch inhibition. Therefore, our proposed model would be as illustrated below (part of the synopsis figure and text)

Fig 5F

Synopsis Figure

5. Figure 3C: The authors should provide the unphosphorylated AKT immunoblot as a control. Same accounts for Figure 3E.

We have included these now.

Fig 3C and E

6. Figure 5B: The levels of NICD overexpression need to be shown.

These constructs are well-established tools in our laboratory, which have been published in previous papers by many laboratories. Since the constructs are fragments of the endogenous proteins, we cannot show the expression on the protein level, as the antibodies cannot detect the remaining protein fragment, because it lacks the Valine at the N-terminus. However, we have provided data showing that the constructs regulate Notch target gene expression on the mRNA level as expected (Fig EV5B, see below). Furthermore, we also validated Notch target gene expression in endothelial cells isolated from skeletal muscle of NICD^{IOE-EC} mice (Fig EV1D, see below). Notably, the induction of Hes and Hey gene expression (Fig. EV1D) is very similar to what we saw in the skeletal muscle endothelial cells from mice for 6 months on HFD (Fig 1A).

Fig EV5B

Fig EV1D

Referee#3 (Remarks for Author):

The authors adequately addressed my concerns.

We thank the reviewer for the positive evaluation of our manuscript.

Referee #4 (Comments on Novelty/Model System for Author):

The performed experiments are well done. The results are novel. The authors show that manipulation of their pathway impacts metabolic health in obesity mouse models.

Referee#4 (Remarks for Author):

The revised version by Jabs et al. has significantly improved. The data are now presented in the appropriate way and the quality of the study is greatly enhanced. However, two critical major questions still remain completely unaddressed in the revised version:

We thank the reviewer for the perceptive comments and for appreciating the work and the data presentation. We really acknowledge the critical inputs regarding metabolic experiments to substantiate the message of this manuscript. We have now performed new experiments to address all of the remaining concerns.

1. Still, there is not a single figure in the paper showing a difference in muscle glucose uptake (neither was insulin uptake itself measured), which would ultimately proof the concept that it is actually insulin-stimulated glucose uptake into muscle that is underlying or at least contributing to the phenotype of the transgenic animals with manipulation of Notch signaling. The in vitro data on caveolin and insulin receptor are very nice, but we don't know whether there are differences in insulin receptor availability in muscle/myocytes actually translating into glucose uptake in vivo. The authors show insulin signaling in muscle and liver but not in other tissues, where insulin needs to cross the barrier. Still, it is possible that the phenotype in vivo is independent of muscle insulin action and muscle glucose uptake and there is a lot of faith required to acknowledge the concept as suggested. Ideally, as I mentioned before, this should be confirmed by clamp studies, in which radiolabeled tracers are used to follow the fate of glucose into the respective tissues. Alternatively, a glucose tolerance test (the authors only assume that postprandial insulin is higher but don't show that - 0 and 10 min blood would be required for that), in which ¹⁴C-deoxyglucose or other tracers are mixed in, would allow to proof their concept. Higher postprandial insulin with lower glucose uptake would prove their concept. Even if they find that this holds also true in other tissues like white or brown fat etc. it would make the study stronger with a broader scope. I do not want to stress this, but

I want to mention that the islet phenotype calls for primary islet insulin release assays, as the differences in vascularization might impact insulin secretion somehow. Showing the impact of Notch signaling on in vivo glucose (insulin?) muscle uptake is imperative for this kind of metabolic study, especially with the strong claims made here.

We agree with the reviewer. We have now performed glucose uptake assays in both Notch loss and gain of function mouse models. We performed a glucose tolerance test with 2-deoxyglucose tracer mixed in. We monitored blood glucose levels at different time points and sampled blood to measure plasma insulin levels (Fig 7, see below). We collected muscle and visceral white adipose tissue (vWAT) samples to measure 2-DG uptake using a commercial kit (CSR-OKP-PMG-K01TE, Cosmo Bio). We found lower glucose uptake in muscle from Notch gain of function mice when compared to controls. This fits perfectly to what we and the reviewer expected. This effect was also accompanied by higher plasma insulin levels during the GTT (Fig 7C and D, see below). Consistently, we found higher glucose uptake in muscle from Notch loss of function mice when compared to controls. We saw a considerable reduction in the plasma insulin levels (Fig 7H and I, see below).

We also analyzed glucose uptake in vWAT as the reviewer asked for a second organ which takes up glucose. However, we did not see any difference in glucose uptake in vWAT (Fig 7E and J, see below). This could very well be attributed to tissue specific differences in endothelial properties between muscles and adipose tissue. There are currently multiple studies ongoing in the vascular biology field to describe all of these organ-specific endothelial properties.

Fig 7

F**G****H****I****J**
We fully understand the reviewer's concern regarding the pancreatic islet phenotype in *Rbpj*^{ΔEC} mice. To address this issue, we performed *ex vivo* glucose stimulated insulin secretion (GSIS) assay in pancreatic islets. There was no difference in the total insulin content in islets isolated from *Rbpj*^{ΔEC} mice compared to littermate controls (Fig EV3E, see below). In addition, insulin secretion after adding 5.6 mM glucose was not altered. Only hyperglycemic conditions led to an increased insulin secretion in this simplified model, most likely due to increased vasculature in islets (Fig EV3F, see below). The higher vascularity could allow for better diffusion of glucose into the pancreatic islets in culture.

Importantly, since insulin secretion *in vivo* is regulated by multiple factors, we subsequently performed an *in vivo* GSIS and measured plasma C-peptide levels after glucose administration in mice. Notably, this did not reveal any significant differences between control and *Rbpj*^{ΔEC} mice (Fig EV3G and H, see below). As such, it is clear that in living mice, there is not excess glucose-mediated insulin secretion in *Rbpj*^{ΔEC} mice.

Considering the slight disparity (only in the severe hyperglycemic condition) in the *ex vivo* vs *in vivo* experiments we have to consider the possibility that there could be an increase in insulin secretion in our *Rbpj*^{ΔEC} mice models under severe hyperglycemic conditions. This could affect the overall systemic glucose homeostasis but it is highly unlikely the only reason for the improvement of systemic glucose homeostasis in *Rbpj*^{ΔEC} mice. This is in particular the case under physiological glucose levels as shown in multiple experiments throughout the manuscript. And also otherwise we would also not see any differences in insulin sensitivity in NICD^{IOE-EC} mice, where there are no differences in the pancreatic islet vasculature.

Fig EV3

2. The paper claims that Notch signaling controls insulin action in muscle, yet there is no physiological regulation of the Notch signaling with fasting and feeding and, therefore, the study does not really have a physiological scope and now has a very different angle. One could argue that Notch signaling constitutively inhibits insulin action through a repression of caveolae formation (why are there no muscle ECs data in Figure 5?). This does not seem to be a developmental defect, and this is an important finding by itself. The authors argue that in the *db/db* model, Notch signaling is enhanced (Why do the authors show lung ECs, and not muscle ECs - in the context of my point #1 brings me back to the question of how much muscle is in the in vivo phenotype or is this systemic - heart we apparently know already...), which might contribute to the insulin resistance in this model somehow. They also show that in the HFD model, inducible deletion of Notch signaling improves glucose tolerance independently of body weight (Fig. 6), which is a very strong experiment and the findings are really exciting. Altogether, this gives this study a more pathophysiological obesity-linked angle. However, there are two major complications (and the in vitro high glucose incubations are not convincing): The *db/db* model a priori is a leptin signaling-deficient model so whether the changes in EC Notch signaling are due to leptin deficiency or due to the metabolic disease in this model cannot not be deduced and this finding remains anecdotal. Much better would be showing muscle ECs from lean and HFD mice next to *db/db* and controls with some stronger readout than qPCR - I think given that this study now has an obesity focus and the authors make very strong claims based on very

little data this is unacceptable. The other is that in the HFD inducible deletion experiment, insulin levels are not shown, ITT was not performed, and muscle glucose uptake was not measured by any means. Altogether, this emphasizes that the authors are not experts in metabolic experiments, but I think their basic concept, particularly as it seems to have some therapeutic benefit for obesity-induced insulin resistance, is still very exciting.

As the reviewer pointed out, our data now show that physiological changes during feeding cycles do not strongly affect Notch signaling in ECs. However, chronic metabolic disturbances such as those seen in obesity, might lead to sustained over-activation of Notch signaling activity in ECs and this subsequently would contribute to impaired insulin sensitivity. As such, it is important to note that the amplitude of Notch activation could differ between physiological and pathological conditions (Kopan, Cold Spring Harbor Perspectives, 2012), and that in particular chronic over-activation affects systemic glucose metabolism.

As the reviewer mentioned, we have now performed new experiments exclusively with endothelial cells freshly isolated from skeletal muscle tissue to replace e.g. data using lung endothelial cells or cardiac muscle endothelial cells. We have included data for caveolar component genes in skeletal muscle endothelial cells in both *Rbpj*^{ΔEC} and *NICD*^{iOE-EC} mice (Fig 5G and H, see below)

Fig 5

As the reviewer suggested, we have now performed experiments on lean (control diet, CD, 10% fat) and high fat diet (HFD, 60% fat) fed obese mice. We also included another group in the cohort where mice were put on high fat and sucrose diet (HFS, 60% fat and 42 g/l sucrose in drinking water *ad libitum*), for a period of 26 weeks starting at 4 weeks of age. We analyzed primary skeletal muscle ECs freshly isolated from these mice. Expression of Notch target genes were elevated in ECs isolated from obese animals (HFD and HFS) compared to ECs derived from CD fed mice (Fig 1A, see below). Notably, the upregulation of *Hes* and *Hey* expression in our *NICD*^{iOE-EC} mice (Fig. EV1D) is very similar to the induction by HFD.

We also analyzed caveolar component gene in skeletal muscle endothelial cells from these mice, which were also changed similar as in the *in vitro* experiments (Fig 5I, see below).

Fig 1A

Fig 5I

In addition, we performed similar analysis with ECs isolated from skeletal muscle of mice on HFD for 3 weeks and 8 weeks. Although, these mice had a notable elevation in their blood glucose levels and body weights, analysis of Notch target genes did not reveal any significant differences (Fig EV1A and B, see below). These results support again the notion that only chronic disturbance of plasma metabolites in obese mice lead to increase in Notch signaling in ECs.

Fig EV1

Regarding using another approach other than qPCR for a readout, we have to mention that we are using freshly isolated primary endothelial cells from mice skeletal tissue without expanding them *in vitro*, as this would not recapitulate the *in vivo* conditions that these cells are exposed to in obese mice. Therefore, we do not have enough starting material to use another approach like Western Blot to detect active Notch signaling as detecting cleaved Notch on a Western Blot requires a lot of input starting material.

We have now included more analysis on HFD *Rbpj*^{ΔEC} mice, as the reviewer had pointed out that we were missing some critical metabolic experiments. We have now performed ITT, glucose uptake assays and measured plasma insulin levels in HFD fed *Rbpj*^{ΔEC} and control mice fed (Fig 7, see below).

Fig 7

We saw an improvement in insulin sensitivity in HFD *Rbpj*^{ΔEC} mice compared to HFD controls. Furthermore, the 2-DG uptake shows improved glucose uptake not only in skeletal muscle but also in vWAT. In addition, the plasma insulin levels were lower both at base line and during GTT in HFD *Rbpj*^{ΔEC} mice.

Taken together, these results clearly indicate that endothelial-specific Notch inhibition improves glucose homeostasis in a HFD induced insulin resistance model.

3rd Editorial Decision

18th Feb 2020

Thank you for the submission of your revised manuscript to EMBO Molecular Medicine. We have now received the enclosed reports from the two referees who were asked to review your revised manuscript. As you will see, they are both supportive of publication, and I am thus pleased to inform you that we will be able to accept your manuscript pending the following final editorial amendments.

***** Reviewer's comments *****

Referee #1 (Remarks for Author):

The authors addressed the majority of issues raised and I have no further comments.

Referee #4 (Comments on Novelty/Model System for Author):

The study provides novel biology with pathophysiological angle, but no "therapeutic impact"

Referee #4 (Remarks for Author):

Thank you for accepting my criticism, and using it to bring this work to an entirely new level of quality – congrats

3rd Revision - authors' response

21st Feb 2020

The authors performed the requested editorial changes.

Corresponding Author Name: Andreas Fischer

Manuscript Number: MM-2018-09271